# GeoBench: Rethinking Multimodal Geometric Problem-Solving via Hierarchical Evaluation

**Yuan Feng**[1,*], **Yue Yang**[1,2,*], **Xiaohan He**[2,3,*], **Jiatong Zhao**[1], **Jianlong Chen**[2,4],
**Daocheng Fu**[2,3], **Qi Liu**[1], **Renqiu Xia**[1,✉], **Bo Zhang**[2], **Junchi Yan**[1,✉]

[1] Shanghai Jiao Tong University   [2] Shanghai Artificial Intelligence Laboratory
[3] Fudan University   [4] The Chinese University of Hong Kong, Shenzhen
* Equal Contribution, ✉ Corresponding Authors

## Abstract

Geometric problem solving constitutes a critical branch of mathematical reasoning, requiring precise analysis of shapes and spatial relationships. Current evaluations of geometric reasoning in vision-language models (VLMs) face limitations, including the risk of test data contamination from textbook-based benchmarks, overemphasis on final answers over reasoning processes, and insufficient diagnostic granularity. To address these issues, we present **GeoBench**, a hierarchical benchmark featuring four reasoning levels in geometric problem-solving: *Visual Perception*, *Goal-Oriented Planning*, *Rigorous Theorem Application*, and *Self-Reflective Backtracking*. Through six formally verified tasks generated via TrustGeoGen, we systematically assess capabilities ranging from attribute extraction to logical error correction. Experiments reveal that while reasoning models like OpenAI-o3 outperform general MLLMs, performance declines significantly with increasing task complexity. Key findings demonstrate that sub-goal decomposition and irrelevant premise filtering critically influence final problem-solving accuracy, whereas Chain-of-Thought prompting unexpectedly degrades performance in some tasks. These findings establish GeoBench as a comprehensive benchmark while offering actionable guidelines for developing geometric problem-solving systems. Our benchmark and code are released at `https://github.com/FrontierX-Lab/GeoBench`.

## 1 Introduction

Geometric Problem Solving (GPS) requires the integration of spatial understanding, theorem application, and logical deduction to analyze shapes, angles, and their relationships (Xia et al., 2024a). Tasks span from identifying basic properties of triangles to constructing complex multi-step proofs, processes that demand rigorous reasoning rather than rote memorization. While recent multi-modal large language models (MLLMs) (Zhang et al., 2024c; Gao et al., 2023) have achieved notable success on geometric tasks, with some even surpassing human performance on benchmarks like GeoQA (Chen et al., 2021), these results obscure critical flaws in evaluation practices that fail to rigorously assess models' holistic geometric reasoning capabilities.

Current benchmarks (Cao & Xiao, 2022; Lu et al., 2021; Zhang et al., 2025; Lu et al., 2024) predominantly rely on problems sourced from public textbooks, creating risks of test data contamination as models exploit memorized patterns rather than true reasoning. Furthermore, existing evaluations focus narrowly on final answers, neglecting the logical processes such as theorem chaining and proof generation that define geometric rigor. Most critically, the lack of diagnostic frameworks leaves unresolved whether failures stem from weak spatial perception, inefficient theorem retrieval, or limited error correction, stalling targeted advancements.

To address these challenges, we propose **GeoBench**, a hierarchical geometric reasoning benchmark containing $1,021$ samples to overcome the limitations of existing geometric reasoning evaluations.

*This work was in part supported by Scientific Research Innovation Capability Support Project for Young Faculty (U40) of the Ministry of Education of China, SRICSPYF-ZY2025019.

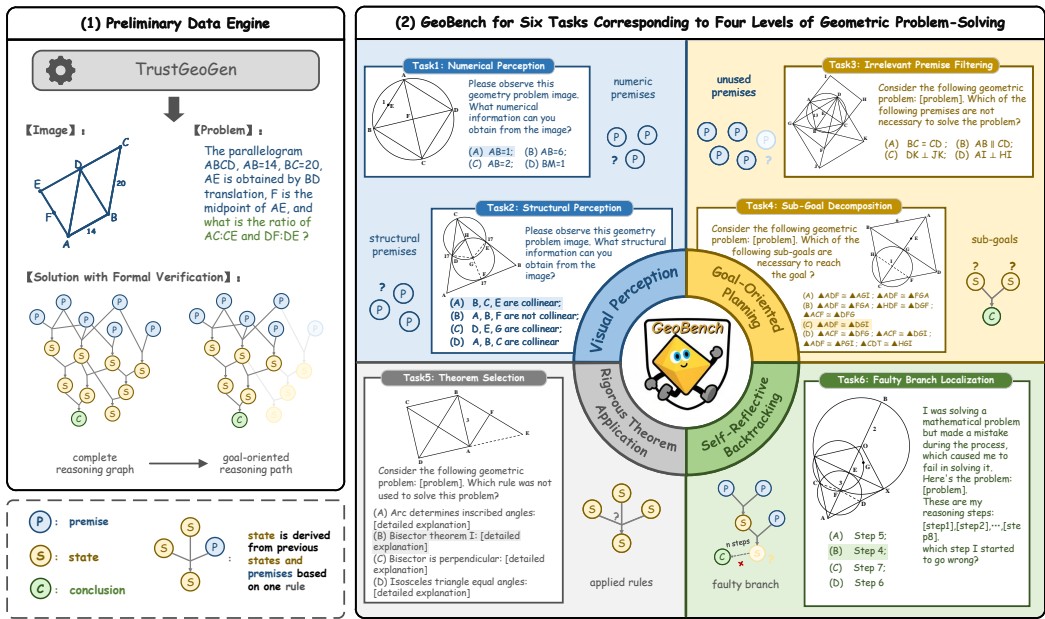

Figure 1: **Overview of GeoBench**: (1) Generation of formally verified geometry problems with images, problems and reasoning graphs via TrustGeoGen (Fu et al., 2025), followed by (2) systematic construction of hierarchical evaluation tasks through four reasoning capability levels.

Grounded in *the van Hiele model of geometric thinking* (Vojkuvkova, 2012), our framework stratifies reasoning into four hierarchical levels: **(1) Visual Perception**, which focuses on extracting numerical and structural details from geometric diagrams; **(2) Goal-Oriented Planning**, requiring decomposing problems into sub-goals and strategic sequencing of steps; **(3) Rigorous Theorem Application**, demanding precise selection of theorems to ensure logical validity; and **(4) Self-Reflective Backtracking**, identifying deviations in reasoning steps and iteratively correcting flawed logic.

Grounded in the proposed four-level evaluation framework, we first generate geometric problems through TrustGeoGen (Fu et al., 2025) with all reasoning steps formally verified via symbolic proof systems. Subsequently, we construct $1,021$ validated samples across six geometric reasoning tasks based on the TrustGeoGen-generated problems and their associated reasoning graphs. As shown in Fig. 1, these tasks systematically operationalize the hierarchical levels: **Numerical Perception** extracts quantitative attributes from diagrams, **Structural Perception** identifies geometric relationships, **Irrelevant Premise Filtering** removes distracting conditions, **Sub-Goal Decomposition** breaks problems into atomic steps, **Theorem Selection** filters inapplicable geometric theorems, and **Faulty Branch Localization** prevents derailment into unproductive reasoning paths. By evaluating models across this hierarchy, we dissect their geometric reasoning capabilities while correlating *final-answer* accuracy with task-specific competencies.

Our experiments evaluated the capabilities of existing general multimodal LLMs and reasoning models on six geometry-related tasks, revealing varying performance levels. A general trend shows that models exhibit declining performance as task difficulty increases, with reasoning models, particularly OpenAI-o3 (OpenAI, 2025), showing the strongest overall capabilities. Furthermore, we investigated the correlation between these six tasks and final problem-solving performance. Experiments on the GeoBench dataset and the OOD GeoQA benchmark indicate that decomposing sub-goals and eliminating irrelevant conditions play more critical roles in solving complex geometry problems, providing insights for enhancing geometric reasoning abilities. Additionally, we examined the impact of Chain-of-Thought (CoT) prompting on geometric task performance in general models. Contrary to common intuition, CoT does not universally improve performance across all tasks, and even reduces effectiveness in self-reflective backtracking tasks. These findings challenge conventional assumptions about CoT's applicability in geometric reasoning scenarios.

**Our contributions are:**

- **Hierarchical Evaluation Framework:** We propose GeoBench, a novel benchmark that systematically evaluates geometric reasoning across four progressive levels, enabling detailed diagnosis of model capabilities beyond answer-only assessments.

- **Key Reasoning Bottlenecks Identification**: Our experiments reveal that sub-goal decomposition and premise filtering critically determine complex problem-solving success, providing actionable insights for improving geometric reasoning models.

- **Task-Specific Prompting Limitations**: We observe that chain-of-thought prompting fails in faulty branch localization tasks, suggesting potential interference when prompts contain misleading reasoning steps, which may divert models from effective error correction.

## 2  RELATED WORK AND PRELIMINARIES

**MLLMs for Geometric Problem-Solving.** Recent advances in Multimodal Large Language Models (MLLMs) (OpenAI, 2024a; 2025; Wu et al., 2024; DeepMind, 2025; Bai et al., 2025) have opened new possibilities for geometric problem solving (GPS). While general MLLMs demonstrate strong performance in vision-language tasks (Li et al., 2023; Xia et al., 2023; 2024b; Yang et al., 2024), their application to GPS remains challenging due to visual-semantic discrepancies, insufficient reasoning capacity, and lack of self-verification mechanisms, etc. Some works address these gaps by data-driven: MAVIS (Zhang et al., 2024c) synthesizes 834K chain-of-thought trajectories to enhance reasoning coherence, while G-LLaVA (Gao et al., 2023) leverages supermodel-guided annotation of 170K geometric solutions, both achieving superior performance over GPT-4o (OpenAI, 2024b) with compact architectures. GeoX (Xia et al., 2024a) pioneers visual-formal language alignment, enabling interpretable theorem verification through symbolic solvers. However, current works overemphasize final answer correctness, calling for developing structured reasoning capacities (i.e., self-reflective reasoning and theorem orchestration) to solve more advanced geometric problems.

**Geometric Problem-Solving Benchmark.** Existing benchmarks primarily collect geometry problems from middle and high school textbooks (Chen et al., 2021; Cao & Xiao, 2022; Lu et al., 2021; Zhang et al., 2023; 2025; Lu et al., 2024; Zhang et al., 2024b). To mitigate annotation challenges and data scarcity, synthetic dataset construction methods have emerged. GeomVerse (Kazemi et al., 2023) enhances authentic problems through LLM-based augmentation, while other approaches (Fu et al., 2025) generate synthetic data via formal language rules to prevent model-induced biases. However, current evaluations focus on final answer accuracy, neglecting deeper analysis of reasoning capabilities. Although GeomRel (Wang et al., 2025) assesses structural diagram comprehension and GeoSense (Xu et al., 2025) examines theorem application patterns, their narrow scopes fail to systematically link subskills to overall problem-solving efficacy. Our GeoBench addresses these limitations by evaluating four critical reasoning levels, as compared with existing benchmarks in Table 1.

**Synthetic Geometry Problems Generation.** TrustGeoGen (Fu et al., 2025) is a scalable, rule-driven formal engine for generating synthetic geometry problems alongside verifiable solutions and structured reasoning pathways. Built on a formal language framework, it integrates three core components: **constructions** (defining geometric elements and premises, e.g., `isosceles triangle ABC` implies AB=AC), **states** (logical propositions about geometric relationships), and **rules** (deductive operations like triangle congruence or parallelism properties). Starting from predefined base scenes (geometric configurations with numerical data), the engine iteratively expands problems by applying constructions to derive initial premises, then progressively generates new states via rule-based inference, forming a traceable reasoning graph from premises to reasoning objectives with corresponding visualizations in Fig. 1.

## 3  METHODOLOGY

This section presents the construction process of GeoBench. In Sec. 3.1, we outline the preparatory work for this benchmark—establishing a complete and correct reasoning chain. Additional details, including the hierarchical evaluation framework, data distribution analysis, and problem difficulty assessment, are presented in Sec. 3.2, 3.3, and 3.4, respectively.

### 3.1  INPUT DATA PREPARATION

For a concrete approach, without loss of generality we use TrustGeoGen (Fu et al., 2025) as our preliminary data engine to obtain geometric *diagrams*, *problems*, and their corresponding *reasoning graphs* through TrustGeoGen as preliminary input. Specifically, TrustGeoGen initiates with a random base scene comprising base premises. Through iterative construction augmentation, the engine constructs complex geometries by expanding the premises, where each newly introduced premise

Table 1: Comparison with Geometric Problem-Solving benchmarks. "F.A." is abbreviated as "Final Answer", explicitly emphasizing the verification of the problem's final result. "V.P.", "G.P.", "R.T.A.", and "S.B." denote "Visual Perception", "Goal-Oriented Planning", "Rigorous Theorem Application", and "Self-Reflective Backtracking" respectively, denoting four-level capability evaluation in GPS.

| Datasets | Size | Language | Level | F.A. | V.P. | G.P. | R.T.A. | S.B. |
|---|---|---|---|---|---|---|---|---|
| GeoQA (Chen et al., 2021) | 754 | EN&CH | Middle School | ✓ | ✗ | ✗ | ✗ | ✗ |
| GeoQA+ (Cao & Xiao, 2022) | 755 | EN&CH | Middle School | ✓ | ✗ | ✗ | ✗ | ✗ |
| Geometry3K (Lu et al., 2021) | 601 | EN | Middle School | ✓ | ✗ | ✗ | ✗ | ✗ |
| PGPS9K (Zhang et al., 2023) | 1,000 | EN | Middle School | ✓ | ✗ | ✗ | ✗ | ✗ |
| GeoEval (Zhang et al., 2025) | 2,000 | EN&CH | Middle&High School | ✓ | ✗ | ✗ | ✗ | ✗ |
| GeomVerse (Kazemi et al., 2023) | 2,000 | EN | Synthetic | ✓ | ✗ | ✗ | ✗ | ✗ |
| MathVista (Lu et al., 2024) | 5,487 | EN | Middle School | ✓ | ✗ | ✗ | ✗ | ✗ |
| MathVerse (Zhang et al., 2024b) | 2,612 | EN | High School | ✓ | ✗ | ✗ | ✗ | ✗ |
| GeomRel (Wang et al., 2025) | 2,629 | EN | Synthetic | ✗ | ✓ | ✗ | ✗ | ✗ |
| GeoSense (Xu et al., 2025) | 1,789 | EN&CH | Synthetic | ✓ | ✗ | ✗ | ✓ | ✗ |
| **GeoBench(ours)** | 1,021 | EN | Synthetic | ✓ | ✓ | ✓ | ✓ | ✓ |

maintains topological consistency with existing elements validated by the compiler (Sicca et al., 2024). The premise set can be defined as $P = \{p_i^r, p_j^n\}$, where $p_i^r$ defines geometric relationships (e.g., `points A B C are collinear`) and $p_j^n$ specifies numerical parameters (e.g., `AB=3`). TrustGeoGen then leverages a predefined set of geometric theorems to infer new states from premises, formulating a complete reasoning graph:

$$\mathcal{G} = (P, S, R, \hookrightarrow) \tag{1}$$

- $P$ represents initial premises from the geometric constructions.
- $S$ denotes the state collection where each element $s \in S$ corresponds to a derived conclusion in reasoning graph.
- $R$ is the set of deductive rules, with each $r \in R$ defines the logic coherence among states.
- $\hookrightarrow \subseteq (S \cup P) \times R \times S = \{(S_r, r, s') \mid S_r \subset (S \cup P), r \in R, s' \in S\}$. The notation $S_r \xhookrightarrow{r} s'$ formalizes the derivation of state $s'$ by applying rule $r$ to states subset $S_r$.

**Goal-oriented Reasoning Path.** To derive the reasoning path for target state $s_t$ (final solution / proof state in geometric problems), TrustGeoGen systematically constructs backward transitions in the complete reasoning graph $\mathcal{G}$ to get the goal-oriented reasoning path:

$$\mathcal{P} = \{(S_{i-1}, r_s, s) \mid \forall s \in S_i, S_{i-1} \xhookrightarrow{r_s} s, i = n, \ldots, 1\} \tag{2}$$

where each triplet $(S_{i-1}, r_s, s_i)$ captures rule application $r_s$ generating state $s_i$ from antecedent states $S_{i-1}$. The process initiates from target state $s_t \in S$ and iteratively traces upstream transitions through rule dependencies until every initial state participating in $\mathcal{G}$ is rigorously encapsulated within the premise set $P$, formally satisfying $S_0 \subseteq P$ where $S_0 := S_{i-1} \mid i = 1$.

## 3.2 HIERARCHICAL EVALUATION FRAMEWORK

Inspired by *the van Hiele model* (Vojkuvkova, 2012), we categorize the geometry problem-solving capabilities into four hierarchical levels comprising six specific tasks. All data are derived from geometry problems, visual diagrams, and corresponding reasoning graphs generated by TrustGeoGen. For each task, we construct answer choices in a multiple-choice format with one correct option and three carefully designed distractors. The developed GeoBench provides a comprehensive evaluation of MLLMs' geometric reasoning abilities through systematically designed question-answer pairs.

### 3.2.1 LEVEL-1: VISUAL PERCEPTION

**Numerical Perception.** We focus on the MLLMs' perception ability of numerical information. Since the initial premise set of a geometric problem is denoted as $P = \{p_i^r, p_j^n\}$, we randomly select one condition from the numerical premise set $p_j^n$, as the ground truth answer. To generate the other three wrong choices, we designed two types of modifications, numerical modification and label modification. Based on a $p_j^n$, a numerical condition is created by randomly altering the value it defines, and a label condition is modified by changing the points into one that does not exist in the geometric scene. For example, if a premise is `AB=6`, a wrong numerical choice could be `AB=4`, and

a wrong label choice can be `AY=9` (point Y does not appear in the image). Therefore, we query the MLLMs with the question: *What numerical condition can you extract from this geometric image?* By providing one correct choice and three wrong choices, we can effectively assess the ability of MLLMs to capture numerical content in geometric images and their potential hallucinations.

**Structural Perception.** To generate the one correct choice, we refer to the geometric scene $P = \{p_i^r, p_j^n\}$ where $p_i^r$ represents geometric relationship conditions. We randomly select one premise $p_i^r$ as the ground truth answer. To generate three wrong choices, we apply inverse negation to the other given relationship conditions. For example, if one $p_i^r$ states "`D, E, F are collinear`", the incorrect choice would be "`D, E, F are not collinear`". In this way, we can construct accurate data to evaluate models' ability to discern geometric structural information. We evaluate the MLLMs' ability to perceive geometric relations in the image by giving it four choices and querying: *What structural information can be extracted from this geometric image?*

### 3.2.2 LEVEL-2: GOAL-ORIENTED PLANNING

**Irrelevant Premise Filtering.** We systematically investigate the capability of MLLMs to filter out redundant premises and extract semantically relevant information in geometric reasoning tasks. Formally, let the initial premise set of a geometric problem be denoted as $P$, with $S_0 \subseteq P$ representing the subset of premises actually employed in the goal-oriented reasoning path. The set of unused premises is consequently defined as $P \setminus S_0$. To ensure meaningful evaluation, we apply two selection criteria to filter out all problems: $P \setminus S_0 \neq \emptyset$ and $|S_0| \geq 3$. The evaluation protocol consists of prompting MLLMs with the interrogative: *Which premise condition is unnecessary for deriving the conclusion?* The ground truth answer is configured as any element $p \in (P \setminus S_0)$, while the distractor set comprises three randomly selected premises $\{p \mid p \in S_0\}_{i=1}^3$.

**Sub-Goal Decomposition.** We aim to investigate the sub-goal decomposition capability of MLLMs. In geometric proof and problem-solving, a common reasoning strategy involves backward chaining, starting from the target conclusion and identifying the necessary preconditions required to derive it. Formally, let $s_t$ represent the target conclusion of a geometric problem, and $r_t \in R$ denote the final inference rule applied to obtain $s_t$. The preceding state set $S_{r_t}$ satisfies the relation $S_{r_t} \xrightarrow{r_t} s_t$, indicating that when the reasoning process reaches state set $S_{r_t}$, the conclusion $s_t$ can be derived through a single application of rule $s_t$. Therefore, we prompt the MLLMs with the question: *What conditions must be established to derive conclusion $s_t$?* Given that $P$ represents the set of premises, we configure the correct option as $S_{r_t} \setminus P$ (the necessary intermediate conditions excluding initial premises). Let $S_{all}$ denote the complete set of intermediate states encountered during the geometric reasoning process. The incorrect options are systematically constructed as $A \cup B$, where $A \subset (S_{r_t} \setminus P)$ and $B \subseteq (S_{all} \setminus P \setminus S_{r_t})$.

### 3.2.3 LEVEL-3: RIGOROUS THEOREM APPLICATION

**Theorem Selection.** We aim to investigate whether MLLMs can effectively identify and filter out unused theorem rules. First, we filter questions where the construction of the inference graph $\mathcal{G}$ involved at least three rules (i.e., $r$). For each filtered question, its corresponding inference process contains a used rules subset $R_{used} = \{r_1, r_2, ..., r_k\}$ with where $k \geq 3$ and $R_{used} \subseteq \mathcal{R}$. Subsequently, we randomly select three distinct rules $\{r_{\text{wrong\_1}}, r_{\text{wrong\_2}}, r_{\text{wrong\_3}}\} \subseteq R_{used}$ from $R_{used}$ as incorrect choices, and randomly select an unused rule $r_{\text{correct}}$ (i.e., $r_{\text{correct}} \notin R_{used}$) from the entire rule repository $\mathcal{R}$ as the correct choice. The final question is a multiple-choice question with four choices, asking to identify *which rule was not used to solve this problem?* The four choices are $r_{\text{wrong\_1}}, r_{\text{wrong\_2}}, r_{\text{wrong\_3}}$, and $r_{\text{correct}}$, where $r_{\text{correct}}$ is the only correct answer.

### 3.2.4 LEVEL-4: SELF-REFLECTIVE BACKTRACKING

**Faulty Branch Localization.** Trace-back reasoning data is defined as a subgraph of the complete reasoning graph $\mathcal{G} = (P, S, R, \hookrightarrow)$, characterized by identical upstream premises and intermediate states but diverging at the final state. As illustrated in Fig. 1, the correct path $\mathcal{P}$ (the goal-oriented reasoning path marked in Fig. 1) consists of the light-colored nodes that successfully reach the correct target state $s_t$. While the subset of light-colored nodes represents a wrong reasoning path $\mathcal{P}_{\text{wrong}}$, although sharing partial upstream nodes with $\mathcal{P}$, it ultimately leads to an erroneous terminal state $s_{\text{wrong}}$. We formally define the **faulty reasoning branch** as the set difference:

$$\mathcal{P}_{\text{faulty}} := \mathcal{P}_{\text{wrong}} \setminus \mathcal{P} \subseteq \mathcal{P}_{\text{wrong}} \tag{3}$$

For localization, we denote $step_0$, the initial divergence step where $\mathcal{P}_{\text{wrong}}$ first deviates from valid reasoning as the correct choice. Similarly, we have also randomly sampled three reasoning steps

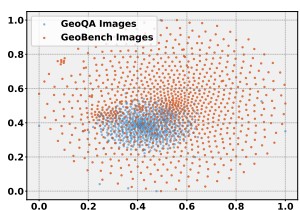 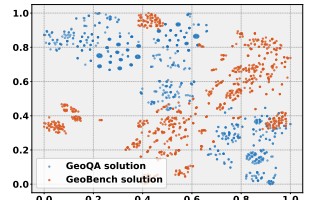 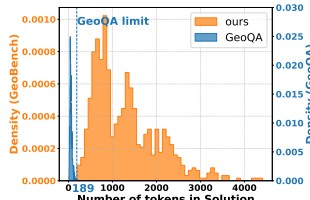

(a) t-SNE of image embeddings     (b) t-SNE of solution embeddings.     (c) Length of solution tokens.

Figure 2: Data Distribution Comparison between GeoBench and GeoQA Chen et al. (2021).

Table 2: Data amount distribution across six tasks in GenBench.

| Total | Numerical Perception | Structural Perception | Irrelevant Premise Filtering | Sub-Goal Decomposition | Theorem Selection | Faulty Branch Localization |
|---|---|---|---|---|---|---|
| **1021** | 168 | 135 | 200 | 200 | 171 | 147 |

from $\mathcal{P}_{wrong}$ to serve as incorrect options. The final task is to prompt the MLLMs to *point out the initial erroneous reasoning step* in a given wrong solution path $\mathcal{P}_{wrong}$ for a geometric problem. Since trace-back reasoning serves as a highly effective approach, the faulty branch localization data provides a reliable inspector to evaluate whether models truly possess self-reflective abilities.

### 3.3 DATA DISTRIBUTION ANALYSIS

**Details about GenBench.** Based on the aforementioned methodology, GenBench incorporates six tasks corresponding to four difficulty levels, comprising a total of $1,021$ question-answer pairs, which aligns with the typical scale of specialized geometric reasoning benchmarks shown in Table 1. The quantitative distribution of these tasks is presented in Table 2. Among the geometric problem-solving objectives, 285 require numerical solutions while the remaining 736 are proof-based. Through TrustGeoGen implementation, we employed 76 geometric constructions, 42 deduction rules, and curated 40 base scenarios, covering most real-world geometric problem scenarios. Additional details are provided in the Appendix C.

**Comparison with OOD Benchmark.** To understand the distributional characteristics of GeoBench, we visualize datasets of images and solutions using t-SNE (Van der Maaten & Hinton, 2008) compared with GeoQA (Chen et al., 2021).

**1) Image-Level.** As shown in Fig. 2a, the **images** embeddings form GeoBench and those from GeoQA form largely disjoint clusters, with our GeoBench being significantly wider spread. This indicates that GeoBench covers a more diverse set of visual pattens, reflecting a broader and more comprehensive range of problem instances.

**2) Solution-Level.** We perform a similar t-SNE(Van der Maaten & Hinton, 2008) analysis on **solutions**, as shown in Fig. 2b, the GeoBench and GeoQA show small overlap. Notably, the GeoBench solutions are more dispersed and span a larger region of the embedding space. Furthermore, we complement this geometric analysis with quantitative comparisons of solution token lengths between the benchmarks. As demonstrated in Fig. 2c, GeoQA solutions exhibit significantly shorter token sequences (maximum 189) compared to GeoBench's solutions, which predominantly maintain thousand-token-scale complexity that underscores GeoBench's heightened solving complexity.

### 3.4 PROBLEMS DIFFICULTY ASSESSMENT

We evaluate geometric problems from GenBench alongside three real-world benchmarks (GeoQA, Geometry3K, and OlympiadBench-Geo (Zhang et al., 2024a)) across multiple reasoning models. Experimental results are presented in Table 3, demonstrating that model performance on GenBench is slightly weaker than that on OlympiadBench-Geo (Olympiad-level), therefore empirically indicating that the complexity and difficulty of synthetic problems in GeoBench have surpassed the middle-school difficulty tiers and generally fall within the elite problem-solving tiers.

## 4 EXPERIMENTS

In this section, we evaluate state-of-the-art multimodal large models on GeoBench and investigate the following research questions through experimental results:

Here, **solution** refers to the solving or proving process targeting geometric objectives, rather than the responses to the six tasks in GeoBench

Table 3: Evaluation results on real-world datasets.

| | GeoQA *(mid-school level)* | Geometry3K *(mid-school level)* | OlympiadBench-Geo *(Olympiad level)* | GeoBench-solving *(ours)* |
|---|---|---|---|---|
| **Qwen2-VL-7b** | 36.34% | 24.46% | 7.14% | 4.17% |
| **GPT-4o** | 42.31% | 31.45% | 13.39% | 22.08% |
| **Gemini-2.5-pro** | 79.58% | 80.70% | 75.00% | 49.58% |

Table 4: Performance of state-of-the-art multi-modal language models on 6 tasks, which are divided into four difficulty levels. "N.P.", "S.P.", "I.P.F.", "S.D.", "T.S.", "F.B.L." correspond to tasks Numerical Perception, Structural Perception, Irrelevant Premise Filtering, Sub-Goal Decomposition, Theorem Selection and Faulty Branch Localization respectively. $\Delta$ denotes accuracy gain versus random choice and ▨/▨ blocks indicate performance above/below random guessing.

| Model | Level 1 | | | | Level 2 | | | | Level 3 | | Level 4 | |
|---|---|---|---|---|---|---|---|---|---|---|---|---|
| | N.P. | | S.P. | | I.P.F | | S.D. | | T.S. | | F.B.L. | |
| | acc | Δ | acc | Δ | acc | Δ | acc | Δ | acc | Δ | acc | Δ |
| Random Choices | 24.64% | - | 25.78% | - | 26.15% | - | 24.55% | - | 25.57% | - | 25.71% | - |
| Human Results | 100.00% | +75.36% | 100.00% | +74.22% | 77.50% | +51.35% | 100.00% | +75.45% | 56.67% | +31.10% | 52.94 % | +27.73% |
| *General MLLMs* | | | | | | | | | | | | |
| LLaVA-1.5-7b [15] | 28.57% | +3.93% | 25.19% | -0.59% | 24.50% | -1.65% | 19.00% | -5.55% | 26.80% | +1.23% | - | - |
| LLaVA-1.5-13b [15] | 25.00% | +0.36% | 24.44% | -1.34% | 30.00% | +3.85% | 32.50% | +7.95% | 26.80% | +1.23% | - | - |
| LLaVA-OV-7b [13] | 58.93% | +34.29% | 22.96% | -2.82% | 26.50% | +0.35% | 33.00% | +8.45% | 16.49% | -9.08% | - | - |
| Molmo [8] | 78.57% | +53.93% | 12.59% | -13.19% | 32.00% | +5.85% | 31.00% | +6.45% | 26.90% | +1.33% | 18.36% | -7.35% |
| Qwen2-VL-7b [26] | 54.17% | +29.53% | 18.52% | -7.26% | 27.00% | +0.85% | 36.00% | +11.45% | 23.71% | -1.86% | 17.69% | -8.02% |
| Qwen2-VL-72b [26] | 86.31% | +61.67% | 29.63% | +3.85% | 31.00% | +4.85% | 60.50% | +35.95% | 37.11% | +11.54% | 21.77% | -3.94% |
| Qwen2.5-VL-7b [2] | **87.50%** | +62.86% | 18.52% | -7.26% | 32.00% | +5.85% | 41.00% | +16.45% | 37.11% | +11.54% | 25.17% | -0.54% |
| Qwen2.5-VL-72b [2] | 85.71% | +61.07% | 40.74% | +14.96% | 38.50% | +12.35% | 77.00% | +52.45% | 47.42% | +21.85% | 26.53% | +0.82% |
| GPT-4o [20] | 66.67% | +42.03% | 22.96% | -2.82% | 44.00% | +17.85% | 57.50% | +32.95% | 35.09% | +9.52% | 23.81% | -1.90% |
| GPT-4V [18] | 51.79% | +27.15% | 34.81% | +9.03% | 35.50% | +9.35% | 57.50% | +32.95% | 38.01% | +12.44% | **27.89%** | +2.18% |
| DeepSeek-VL2 [30] | 66.07% | +41.43% | 25.19% | -0.59% | 27.00% | +0.85% | 36.00% | +11.45% | 23.98% | -1.59% | 22.45% | -3.26% |
| *Reasoning MLLMs* | | | | | | | | | | | | |
| QvQ-72b [22] | 61.31% | +36.67% | 27.41% | +1.63% | 29.50% | +3.35% | 59.00% | +34.45% | 44.33% | +18.76% | 18.37% | -7.34% |
| OpenAI-o1 [19] | 75.00% | +50.36% | 65.19% | +39.41% | 61.50% | +35.35% | 77.00% | +52.45% | 53.22% | +27.65% | **27.89%** | +2.18% |
| OpenAI-o3 [21] | 80.95% | +56.31% | **74.81%** | +49.03% | 70.00% | +43.85% | **91.00%** | +66.45% | **54.39%** | +28.82% | 22.45% | -3.26% |
| Claude-3.7-sonnet [1] | **87.50%** | +62.86% | 46.67% | +20.89% | 52.50% | +26.35% | 67.50% | +42.95% | 45.03% | +19.46% | 21.77% | -3.94% |
| Gemini-2.5-pro [6] | 80.95% | +56.31% | 60.00% | +34.22% | **74.00%** | +47.85% | 87.00% | +62.45% | 45.03% | +19.46% | 18.37% | -7.34% |
| DeepSeek-R1 [7] | - | - | - | - | 57.50% | +31.35% | 82.00% | +57.45% | 41.52% | +15.95% | 21.77% | -3.94% |
| *Agent* | | | | | | | | | | | | |
| CodePlot [9] | 73.81% | +49.17% | 38.52% | +12.74% | 30.50% | +4.35% | 41.50% | +16.95% | 19.30% | -6.27% | 15.65% | -10.06% |

- *How do current MLLMs perform in geometric reasoning across different hierarchical levels?*
- *Which geometric reasoning abilities are more indicative of a multimodal model's problem-solving proficiency in geometry?*
- *Can the evaluation results from GeoBench generalize to OOD geometric test sets?*
- *Is Chain of Thought (CoT) truly effective for geometric reasoning tasks?*

## 4.1 DATASETS, METRICS AND IMPLEMENTATION DETAILS

**Datasets.** As written in Sec. 3, we divide geometric reasoning capabilities into four levels and use TrustGeoGen to generate six specialized geometric tasks covering all levels. Each instance consists of a multiple-choice question, including a natural language problem description, one correct option, and three incorrect options. Regarding answer validation, we confirm that our dedicated team of eight doctoral researchers and PhD candidates, all possessing strong mathematical competencies, performed comprehensive cross-verification.

**Metrics.** For General MLLMs, we adjust the prompt to have models output option indices directly. For Reasoning MLLMs, which generate reasoning processes, we specify an output format in the prompt that requires placing the deduced option index within \boxed{} brackets. After extracting answers, we calculate the accuracy($acc$) for each task and compare it with the baseline model (random choices) to derive relative performance difference($\Delta$).

**Implementation Details.** We conducted all inference evaluations of open-source models on 8 NVIDIA A100 (80G) GPUs, while performing evaluations of proprietary models on CPUs. For detailed parameter configuration, please refer to Appendix C.1.

## 4.2 HUMAN EVALUATION

We equally distributed the data to five human experts (each holding a Ph.D. from a top 1% national university) for manual evaluation. The results, calculated as the arithmetic mean with the exclusion of the highest and lowest ratings, are presented in the Table 4. Human experts consistently achieved

---

LLaVA-1.5 series models cannot be evaluated on the F.B.L. task due to input token length limitations.

Since DeepSeek-R1 is a text-only model, it cannot be tested on Level 1 visual tasks. For all other tasks, visual conditions are fully provided in textual descriptions.

Table 5: Comparative performance state-of-the-art multi-modal language models on 6 tasks in GeoBench, with final answer accuracy on GeoBench-solving, GeoQA and Geometry3K datasets.

| Model | Level 1 | | Level 2 | | Level 3 | Level 4 | GeoBench-solving | GeoQA | Geometry3K |
|---|---|---|---|---|---|---|---|---|---|
| | N.P. | S.P. | I.P.F. | S.D. | T.S. | F.B.L. | | | |
| *General MLLMs* | | | | | | | | | |
| LLaVa-1.5-7b [15] | 28.57% | 25.19% | 24.50% | 19.00% | 26.80% | - | 2.50% | 17.11% | 1.16% |
| LLaVa-1.5-13b [15] | 25.00% | 24.44% | 30.00% | 32.50% | 26.80% | - | 3.75% | 18.57% | 2.33% |
| Qwen2-VL-7b [26] | 54.17% | 18.52% | 27.00% | 36.00% | 23.71% | 17.69% | 4.17% | 36.34% | 24.46% |
| Qwen2-VL-72b [26] | 86.31% | 29.63% | 31.00% | 60.50% | 37.11% | 21.77% | 10.83% | 55.70% | 34.44% |
| Qwen2.5-VL-7b [2] | 87.50% | 18.52% | 32.00% | 41.00% | 37.11% | 25.17% | 11.67% | 54.64% | 35.94% |
| Qwen2.5-VL-72b [2] | 85.71% | 40.74% | 38.50% | 77.00% | 47.42% | 26.53% | 17.50% | 67.90% | 48.42% |
| GPT-4o [20] | 66.67% | 22.96% | 44.00% | 57.50% | 35.09% | 23.81% | 22.08% | 42.31% | 31.45% |
| GPT-4v [29] | 51.79% | 34.81% | 35.50% | 57.50% | 38.01% | 27.89% | 19.17% | 35.94% | 20.63% |
| *Reasoning MLLMs* | | | | | | | | | |
| QvQ-72b [22] | 61.31% | 27.41% | 29.50% | 59.00% | 44.33% | 18.37% | - | 61.67% | 45.09% |
| OpenAI-o1 [19] | 75.00% | 65.19% | 61.50% | 77.00% | 53.22% | 27.89% | 48.33% | 75.83% | 71.29% |
| OpenAI-o3 [21] | 80.95% | 74.81% | 70.00% | 91.00% | 54.39% | 22.45% | 53.33% | 83.33% | 80.20% |
| Gemini-2.5-pro [6] | 80.95% | 60.00% | 74.00% | 87.00% | 45.03% | 18.37% | 49.58% | 79.58% | 80.70% |
| Claude-3.7-Sonnet [1] | 87.50% | 46.67% | 52.50% | 67.50% | 45.03% | 21.77% | 35.42% | 49.73% | 33.28% |

higher accuracy than MLLMs across all tasks. This performance gap was particularly pronounced for the N.P. and S.P. tasks, which require image-grounding, and the F.B.L. task, which involves more complex reasoning.

## 4.3 PERFORMANCE ON ALL 4 LEVELS

In Table 4, the results on the GeoBench reveal that MLLMs possess a certain level of reasoning capability in solving geometric problems, with performance generally declining as the difficulty level increases. Among the existing open-source models, the evaluation results of the LLaVa-1.5 (Liu et al., 2024) series are comparable to random selection, indicating relatively weak geometric perception and reasoning abilities. The LLaVa-OV-7b (Li et al., 2024) model only slightly outperforms others in Level 1 (Visual Perception) tests. Other models demonstrate strong performance in Level 1 and Level 2 (Goal-Oriented Planning), with modest improvements over baseline models in Level 3 (Rigorous Theorem Application). Comparative analysis suggests that as model parameters increase and versions advance, genuine geometric reasoning capabilities also improve.

We included the Molmo (Deitke et al., 2024) model (an open-source vision-language model trained on the PixMo (Deitke et al., 2024) dataset, a collection of image-text pairs) in our testing, with the results presented in the Table 4. Based on the results presented in the table, it can be observed that the performance of the Molmo (Deitke et al., 2024) model falls short of that achieved by proprietary commercial MLLMs models, indicating its relatively limited capability in geometric reasoning.

Proprietary commercial models demonstrate superior reasoning capabilities. For instance, OpenAI-o3 (OpenAI, 2025) achieves 91.00% accuracy on the S.D. task and 54.39% accuracy on the T.S. task, while OpenAI-o1 also attains a comparable accuracy of 53.22% on the T.S. task—significantly outperforming most other models which generally exhibit poor performance on these tasks.

Across most tasks, reasoning MLLMs outperform general MLLMs. Top-performing models such as OpenAI-o1 (OpenAI, 2024a), OpenAI-o3 (OpenAI, 2025), Claude-3.7-sonnet (Anthropic, 2024), and Gemini-2.5-pro (DeepMind, 2025) surpass 70% of general models in N.P. task, exceed all general models in S.P. and I.P.F. tasks, and outperform over 90% of general models in S.D. and T.S. tasks. All evaluated MLLMs exhibited severe limitations in F.B.L. performance. The highest accuracy achieved by OpenAI-o1 (OpenAI, 2024a) reached merely 27.89%, a statistically insignificant improvement compared to the 25.71% baseline of random choices.

Finally, we also selected an agent called CodePlot (Duan et al., 2025) for testing, which can use Python to generate plots during the reasoning process to assist in thinking. The experimental results show that its performance is somewhat inferior to that of commercial large models, particularly with a significant disadvantage in T.S. and F.B.L. tasks.

We conducted repeated experiments to demonstrate the reliability of the results. For details, please refer to the Appendix E.

Table 6: Spearman correlation ($\rho$) among six tasks and final-answer evaluation.

| | N.P. | S.P. | I.P.F. | S.D. | T.S. | F.B.L. |
|---|---|---|---|---|---|---|
| with GeoBench-Solving | 0.40351 | 0.75657 | 0.97902 | 0.88772 | 0.82954 | 0.49561 |
| with GeoQA | 0.65859 | 0.67181 | 0.74945 | 0.93392 | 0.84896 | 0.50057 |
| with Geometry3K | 0.59780 | 0.64374 | 0.71429 | 0.88981 | 0.81932 | 0.38122 |

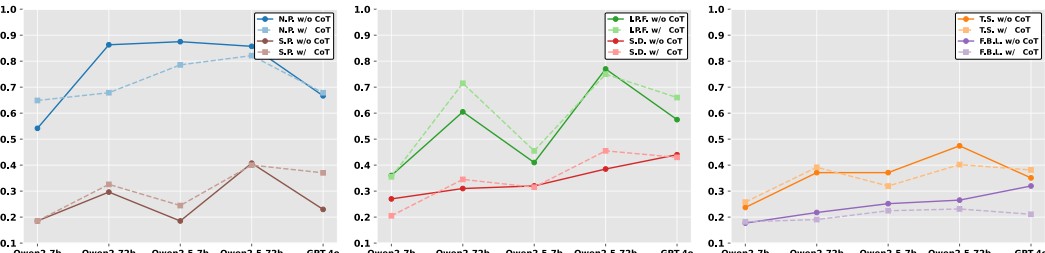

Figure 3: Performance comparison w/ & w/o Chain-of-Thought (CoT) across six tasks in GeoBench.

## 4.4 CORRELATION AMONG TASK-SPECIFIC CAPABILITIES AND FINAL-ANSWER PERFORMANCE

To investigate which geometric reasoning abilities have a greater impact on the performance of MLLMs in solving geometric problems, as detailed in Table 5, we validate the final answer on *GeoBench-solving*: the preliminary set of 285 geometric problem-solving questions with numeric answer from GeoBench. For each task, we constructed a feature vector $X_i$ based on the test results across different MLLMs and computed the Spearman Correlation Coefficient (De Winter et al., 2016) between $X_i$ and the GeoBench-solving vector $Y$ to analyze the influence of each task on GPS ability.

The Spearman Correlation Coefficient ($\rho$) is a rank-based measure that indicates the direction and strength of the relationship between an independent variable $X$ and a dependent variable $Y$. Its value ranges from $[-1, 1]$, where values closer to 1 or $-1$ indicate a stronger correlation. As observed in Table 6, the first six tasks exhibit a positive correlation with GeoBench-solving results, meaning that stronger geometric reasoning abilities correspond to better problem-solving performance. Among them, task I.P.F., S.D., and T.S. show the strongest correlations, followed by S.P. and N.P., with F.R.L. having the weakest correlation. This finding confirms that **Sub-Goal Decomposition**, **Irrelevant Premise Filtering**, and **Theorem Selection** abilities significantly influence MLLMs' geometric problem-solving capabilities.

## 4.5 ANALYSIS ON OUT-OF-DISTRIBUTION DATASET

To validate the out-of-distribution (OOD) generalization ability, we evaluated the geometric problem-solving performance of MLLMs on GeoQA and Geometry3K (see Table 5) and further computed the Spearman correlation coefficients between each task and the GeoQA, Geometry3K evaluation results (in Table 6). The correlation analysis reveals that task S.D. and T.S. exhibit the strongest correlations, followed by I.P.F., then N.P. and S.P., with F.B.L. showing the weakest correlation. This trend remains consistent across GeoQA, Geometry3K, and GeoBench-solving, collectively confirming the robust out-of-distribution generalization capability intrinsic to GeoBench. By leveraging our benchmark, we confirm that the improvement in MLLMs' geometric reasoning abilities is genuine and not merely a result of overfitting to the answer patterns in GeoQA or Geometry3K.

## 4.6 APPLICABILITY OF CHAIN-OF-THOUGHT PROMPTING

To enhance the reasoning capabilities of General MLLMs, we explored the optimization of prompts using Chain-of-Thought (CoT) (Wei et al., 2022) methodology, which yielded intriguing findings. Specifically, we conducted ablation studies on the Qwen series (Wang et al., 2024; Bai et al., 2025) of large models as well as GPT-4o (OpenAI, 2024b), evaluating their performance across six tasks with both CoT-enhanced prompts and non-CoT prompts. In the CoT condition, models were prompted with the instruction `"let's think step by step"`, which elicited detailed reasoning processes from the models. Conversely, in the non-CoT condition, the prompt `"only output the answer"` was employed, restricting the models to provide answers without explicit reasoning steps. Additional details regarding the prompt construction and implementation are provided in Appendix D.

Table 7: Performance on Perturbed Dataset.

| | Level 1 | | Level 2 | | Level 3 | Level 4 |
|---|---|---|---|---|---|---|
| | N.P. | S.P. | I.P.F. | S.D. | T.S. | F.B.L. |
| Qwen2-VL-7b [26] | 57.14% | 23.70% | 25.50% | 38.00% | 26.32% | 22.45% |
| Qwen2-VL-72b [26] | 80.95% | 33.34% | 28.00% | 56.50% | 39.77% | 19.73% |
| Qwen2.5-VL-7b [2] | 79.17% | 20.74% | 28.50% | 35.50% | 35.09% | 23.13% |
| Qwen2.5-VL-72b [2] | 83.93% | 40.00% | 39.00% | 76.00% | 44.44% | 24.49% |

Table 8: Text-Only Experiments Results.

| | Level 1 | | Level 2 | | Level 3 | Level 4 |
|---|---|---|---|---|---|---|
| | N.P. | S.P. | I.P.F. | S.D. | T.S. | F.B.L. |
| Qwen2-VL-7b [26] w/ text | 33.93% | 23.70% | 29.50% | 36.00% | 23.71% | 22.45% |
| Qwen2-VL-7b [26] w/ text+image | 54.17% | 18.52% | 27.00% | 36.00% | 23.71% | 17.69% |
| Qwen2-VL-72b [26] w/ text | 47.62% | 19.26% | 27.50% | 59.50% | 30.99% | 21.09% |
| Qwen2-VL-72b [26] w/ text+image | 86.31% | 29.63% | 31.00% | 60.50% | 37.11% | 21.77% |
| Qwen2.5-VL-7b [2] w/ text | 39.29% | 19.26% | 28.00% | 40.00% | 28.65% | 24.49% |
| Qwen2.5-VL-7b [2] w/ text+image | 87.50% | 18.52% | 32.00% | 41.00% | 37.11% | 25.17% |
| Qwen2.5-VL-72b [2] w/ text | 51.19% | 22.96% | 39.00% | 75.00% | 43.27% | 21.77% |
| Qwen2.5-VL-72b [2] w/ text+image | 85.71% | 40.74% | 38.50% | 77.00% | 47.42% | 26.53% |

The experimental results are visualized in the line chart presented in Fig. 3. Analysis of the trend lines reveals that the CoT approach demonstrates marginally superior performance compared to the non-CoT method in tasks S.P., I.P.F., and S.D., while showing inconsistent results in N.P. and T.S. tasks. Notably, the results of F.B.L. task indicate a systematic underperformance of the CoT method relative to its non-CoT counterpart, suggesting that CoT is ineffective for this particular task - specifically, it fails to enhance the models' Faulty Branch Localization capability.

Upon closer examination, we hypothesize that this phenomenon may be attributed to the presence of erroneous reasoning chains in F.B.L. task prompts. These misleading cues could potentially interfere with the reasoning processes of general MLLMs when handling complex geometric problem-solving tasks. This observation suggests a nuanced perspective: While CoT can enhance model performance under certain conditions, its effectiveness for geometric reasoning appears contingent on information reliability. Specifically in complex geometric scenarios, our findings indicate that CoT methodology might not consistently improve performance across general MLLMs, revealing important limitations regarding its application in mathematically rigorous domains.

### 4.7 PERTURBED DATASET EXPERIMENTS

We applied random rotations of $\pm 20°$ to the images to create a perturbed test set and conducted corresponding experiments. The results are presented in the Table 7 below. Based on a comparative analysis with the evaluation results of the Qwen (Wang et al., 2024; Bai et al., 2025) MLLMs presented in Table 4 of the paper, we observe that the performance metrics remain remarkably consistent. This finding empirically demonstrates that minor image perturbations (within $\pm 20°$ rotation) do not significantly affect the evaluation outcomes.

### 4.8 TEXT-ONLY EXPERIMENTS

We conducted text-only ablation experiments on MLLMs, with results presented in the Table 8. Experimental findings indicate that most vision-language models exhibit inferior evaluation performance when processing text-only inputs compared to multimodal inputs incorporating both images and text.

### 5 CONCLUSION

Our work establishes GeoBench as a diagnostic framework for rigorously evaluating geometric reasoning through four hierarchical capability levels. Experimental analyses on MLLMs reveal two critical bottlenecks: Sub-Goal Decomposition and Irrelevant Premise Filtering significantly impact complex problem-solving success, while Chain-of-Thought prompting exhibits task-specific limitations, particularly impairing Self-Reflective Backtracking. These findings challenge the assumption of universal CoT efficacy in geometric reasoning and highlight the necessity of structured evaluation beyond final-answer metrics. The GeoBench framework provides actionable pathways for advancing geometric reasoning through targeted capability enhancement. Future work should expand task diversity while rigorously refining structured evaluation protocols for mathematical precision.

**Ethics Statement.** Our proposed GeoBench may ultimately find applications in the field of educational technology, such as in intelligent tutoring systems. We are committed to advancing this work responsibly, with the objective of assisting human learning and fostering interest in mathematics, rather than replacing deep human reasoning and creative thinking. We will refrain from pursuing any applications that could potentially facilitate academic misconduct, such as automated problem-solving for cheating purposes. Besides, we will avoid using language that might lead to public misunderstanding or excessive hype (claims like "AI has surpassed human mathematical abilities"). Instead, we will accurately and objectively delineate the MLLMs' capabilities, emphasizing that its current performance is primarily evaluated within specific, controlled synthetic environments.

**Reproducibility Statement.** The complete source code and the full evaluation dataset have been made publicly available at the anonymous link: `https://github.com/FrontierX-Lab/GeoBench`. Furthermore, both the code and the data will be open-sourced in the future.

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

## A    DECLARATION OF AI USE

We utilized large language models, including DeepSeek-V3, GPT-4, and Gemini, to assist with the writing process by:

- Translating technical terms into idiomatic English.

- Correcting grammatical errors and improving sentence fluency.

- Polishing the overall wording.

We assure that ideas, methods, code implementations, experiments, analyses, and conclusions are done by human researchers ourselves.

## B    LIMITATIONS AND BROADER IMPACT

### B.1    FALIURE CASES STUDY

We conducted a failure analysis of GeoBench across its six tasks to identify key limitations. In the N.P. and S.P. tasks, complex diagrams sometimes caused perceptual errors in large models, such as misreading labeled line segments or overlooking point collinearity (Fig. 4a, 4b). For I.P.F., models produced incorrect intermediate conclusions or misinterpreted problem descriptions (Fig. 4c). In S.D., they often mistook irrelevant premises as necessary and introduced hallucinated conditions (Fig. 4d). During T.S., models drew erroneous conclusions from incomplete evidence (Fig. 4e). Notably, in F.B.L., MLLMs primarily detected *factual errors* (incorrect single-step deductions) rather than the intended *directional errors* (steps deviating from the goal) (Fig. 4f).

### B.2    LIMITATIONS

In this work, we propose GeoBench, a novel benchmark that systematically evaluates geometric reasoning across four progressive levels. Currently, our geoBench is only applicable to planar geometry, as the utilized data engine TrustGeoGen and generation pipeline are designed specifically for 2D geometric problems. To extend the benchmark to 3D geometric structures, new spatial rules and formal definitions tailored to three-dimensional settings would need to be introduced, marking a promising direction for future research.

### B.3    BROADER IMPACT

GeoBench aims to reshape the evaluation paradigm for geometric reasoning in large multimodal language models by shifting the focus from answer correctness to reasoning fidelity. This benchmark provides a standardized, verifiable framework for diagnosing fine-grained reasoning failures, which may inform the design of educational technologies, tutoring systems, and trustworthy models in mathematics. Furthermore, our task decomposition reveals the intelligence demands required at different levels of geometric abstraction, offering insights for geometric reasoning improvement and mathematical curriculum design.

Importantly, we recognize existing score-centric benchmarks could risk overfitting models to benchmark-specific reasoning templates, thereby reinforcing narrow optimization rather than general geometric understanding. To mitigate this, we propose the multi-dimensional GeoBench, advocating that future applications should prioritize diversity, such as image perception, problem formulations, and reasoning paths. Ultimately, GeoBench contributes toward building more interpretable and cognitively aligned models capable of solving geometry problems rigorously and systematically.

## C    EXPERIMENTAL DETAILS

### C.1    PARAMETER CONFIGURATION

For all MLLMs, we set the parameters as `temperature = 0.95`, `max_tokens = 8192`, and `top_p = 0.7`. For API-based models (including OpenAI-o1/o3, GPT-4o/4v, DeepSeek-R1, Claude-3.7-sonnet, and Gemini-2.5-pro), we implemented a retry logic that allows up to 5 API reattempts in case of network connection errors.

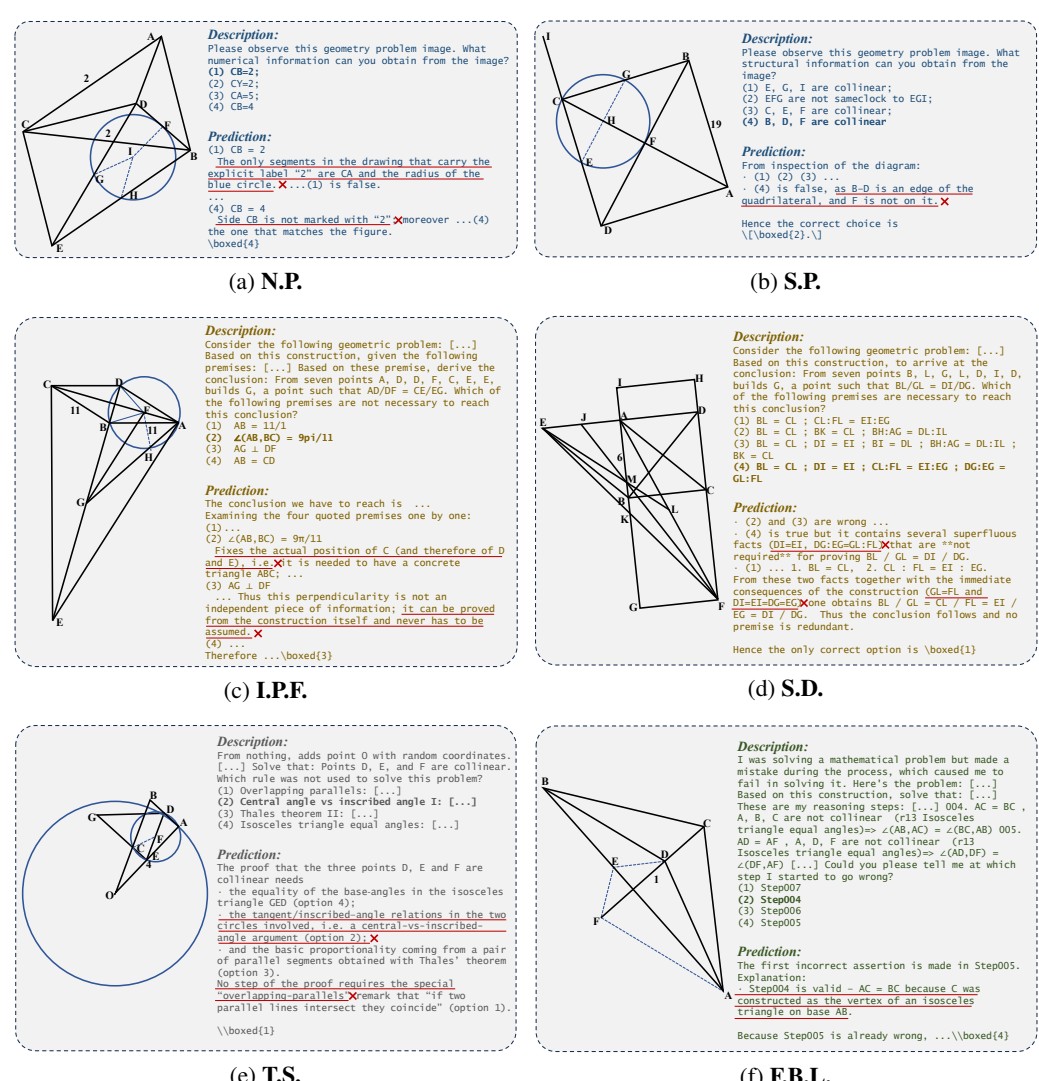

Figure 4: Failure cases (some question descriptions and prediction content omitted for space limit).

## C.2 EVALUATION PROMPT

In this section, we will provide detailed prompt templates along with examples. Fig. 5 illustrates the basic structure of the prompt template, which consists of the following components: 1) Problem NL; 2) Question; 3) Multiple-choice task description; 4) Options; and 5) Output instructions.

Here, `Problem NL` refers to the natural language description of the problem. The output instructions are further divided into two variants: with and without Chain-of-Thought (CoT). Fig. 5a demonstrates the prompt without CoT, while Fig. 5b presents the prompt incorporating CoT.

## C.3 MODEL NAME

In this section, we provide a list of model names corresponding to selected MLLMs that employ API calls for inference, as shown in Table 9.

## C.4 ANSWER EXTRACTION

To compute the multiple-choice question accuracy(*acc*) of MLLMs, it is necessary to extract the model's predicted answers from its reasoning outputs for comparison with the ground truth.

```
<Problem NL> + <Question> +
"The following will present four numerically
labeled options, including one correct option
and three incorrect options. Please verify the
contents of options (1) to (4) in sequence to
determine whether they meet the problem's
requirements, i.e., whether they are correct.
Finally, output the serial number of the
correct option.

The options are as follows:
(1) ...
(2) ...
(3) ...
(4) ...

When outputting, only provide the serial number
without the text following it. Represent the
serial number using parentheses + a digit, e.g.,
(1), (3), etc.

Please note, only output the answer, without
any thought process or additional information."
```

(a) w/o CoT

```
<Problem NL> + <Question> +
"The following will present four numerically
labeled options, including one correct option
and three incorrect options. Please verify the
contents of options (1) to (4) in sequence to
determine whether they meet the problem's
requirements, i.e., whether they are correct.
Finally, output the serial number of the
correct option.

The options are as follows:
(1) ...
(2) ...
(3) ...
(4) ...

Please think step by step, output the reasoning
process, and select the single correct answer.
Represent the answer as a number in parentheses
(1-4) and enclose it in \boxed{}. Note that the
correct answer must be a single choice between
1 and 4; do not output more or fewer answers,
and do not output 'None' as an answer. For
example, ...\boxed{(2)}, ...\boxed{(4)}, etc."
```

(b) w/ CoT

Figure 5: Prompts Format

Table 9: Model Names for MLLMs employing API Calls

| Model | Model Name |
|---|---|
| GPT-4o | gpt-4o |
| GPT-4V | gpt-4-1106-preview |
| OpenAI-o1 | o1-2024-12-17 |
| OpenAI-o3 | o3-2025-04-16 |
| Claude-3.7-sonnet | claude-3-7-sonnet-20250219 |
| Gemini-2.5-pro | gemini-2.5-pro-exp-03-25 |
| DeepSeek-R1 | deepseek-r1 |

For General MLLMs not employing the CoT approach, we instruct them to output only the answer choice following the prompt template in Sec. C.2. Empirical observations confirm that these models consistently comply by generating only a single option identifier. We therefore directly compare their raw outputs with the ground-truth labels for accuracy evaluation.

For Reasoning MLLMs or General MLLMs utilizing the CoT method, the model outputs a complete reasoning process before generating the final answer. Therefore, we first employ the large language model to extract the answer from the reasoning output, then match substrings in the form of `(1)`, `(2)`, etc., and compare them with the ground-truth labels to compute accuracy.

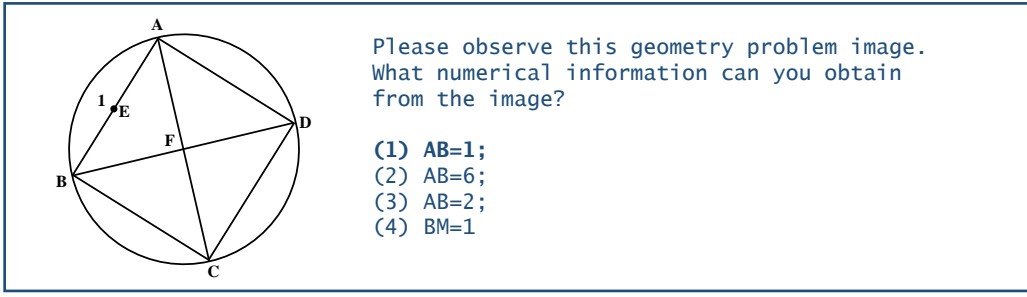

(a) Numerical Perception

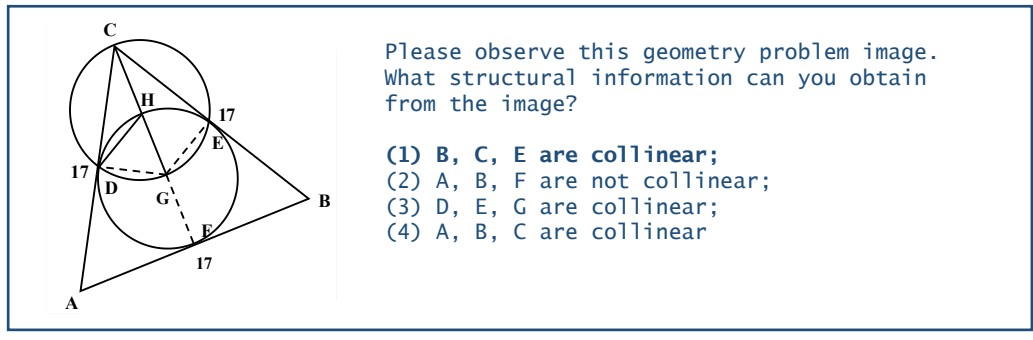

(b) Structural Perception

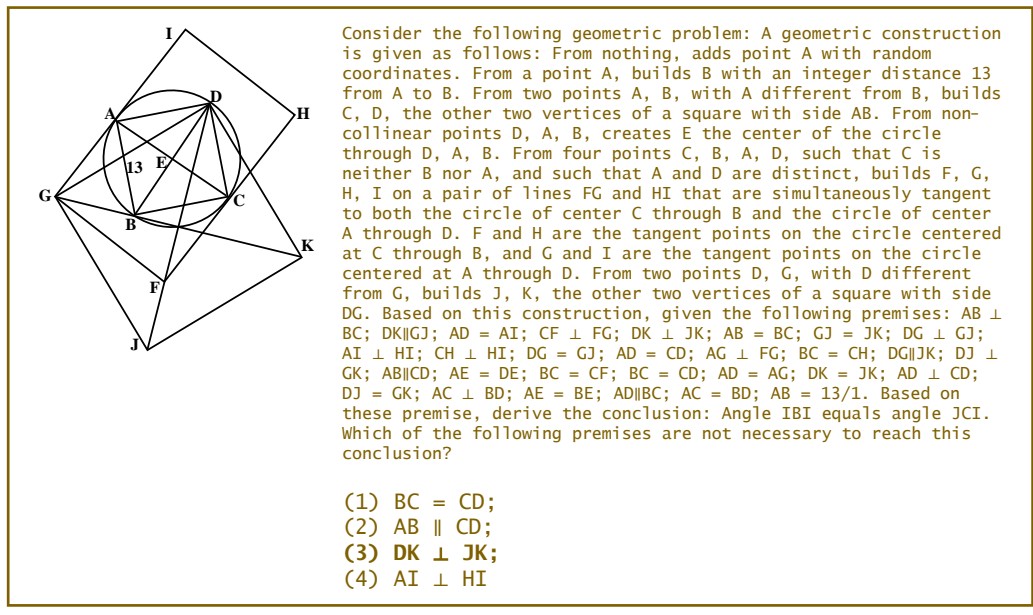

(c) Irrelevant Premise Filtering

Figure 6: Visualization Examples

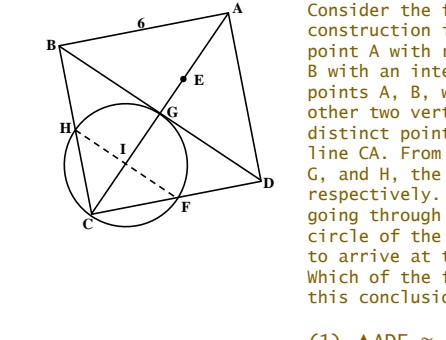

Consider the following geometric problem: A geometric construction is given as follows: From nothing, adds point A with random coordinates. From a point A, builds B with an integer distance 6 from A to B. From two points A, B, with A different from B, builds C, D, the other two vertices of a square with side AB. From two distinct points C, A, builds E, another point on the line CA. From three non-collinear points B, C, D, adds F, G, and H, the midpoints of sides CD, BD, and BC, respectively. It also adds I, the center of the circle going through F, G, and H, which is also the nine points circle of the triangle BCD. Based on this construction, to arrive at the conclusion: Angle FDF equals angle IDI. Which of the following premises are necessary to reach this conclusion?

(1)  ▲ADF ≅ ▲AGI ;  ▲ADF ≅ ▲FGA
(2)  ▲ADF ≅ ▲FGA ;  ▲HDF ≅ ▲DGF ;  ▲ACF ≅ ▲DFG
**(3)  ▲ADF ≅ ▲DGI**
(4)  ▲ACF ≅ ▲DFG ;  ▲ACF ≅ ▲DGI ;  ▲ADF ≅ ▲PGI ; ▲CDT ≅ ▲HGI

(a) Sub-Goal Decomposition

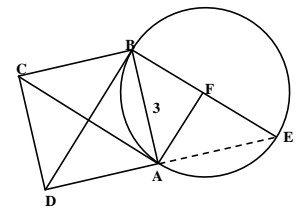

Consider the following geometric problem: From nothing, adds point A with random coordinates. From a point A, builds B with an integer distance 3 from A to B. From three non-collinear points A, B, and D, adds E, the other intersection of the circle of center A through D and the circle of center B through D. From non-collinear points E, A, B, creates F the center of the circle through E, A, B. Solve that: Angle CAD equals angle EAF Which rule was not used to solve this problem?

(1) Arc determines inscribed angles (tangent): This rule corresponds to r03 in the case the arc is determined by a tangent line. An inscribed angle determining that same arc will be congruent to the angle determining that arc with one leg being the tangent line at the vertex of the arc.
**(2) Bisector theorem I: One direction of the bisector theorem: if a line through a vertex of a triangle cuts the opposite side into two segments that are in proportion as the neighboring sides of the triangle, the line bisects the angle at the vertex it cuts.**
(3) Bisector is perpendicular: This rule is the reverse direction of r22. It says that the locus of the points that are equidistant to the two vertices of a segment AB is a straight line perpendicular to AB.
(4) Isosceles triangle equal angles: The theorem says that the base angles of an isosceles triangle are congruent.

(b) Theorem Selection

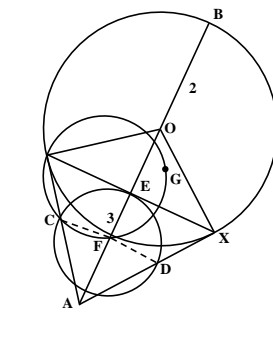

I was solving a mathematical problem but made a mistake during the process, which caused me to fail in solving it. Here's the problem: From nothing, adds point O with random coordinates. From a point O, builds B with an integer distance 2 from O to B. From a point O, builds A with an integer distance 3 from O to A. From three different points A, B, O, builds X and Y, the points of tangency of the two lines through A tangent to the circle of center O through B. From three non-collinear points X, Y, A, adds C, D, and E, the midpoints of sides YA, XA, and XY, respectively. It also adds F, the center of the circle going through C, D, and E, which is also the nine points circle of the triangle XYA. From three non-collinear points Y, F, and C, builds G a point on the circle through Y, F, and C. Based on this construction, solve that: From seven points C, X, F, X, D, Y, F, builds Y, a point such that CX/FX = DY/FY. These are my reasoning steps: BO = OX , BO = 2/1 (Ratio Chasing)=> OX = 2/1 001. AO = 3/1 , OX = 2/1 , AX ⊥ OX (Pythagoras Verification)=> Pythagorean Theorem's premises on X, A, O are satisfied 002. Pythagorean Theorem's premises on X, A, O are satisfied (r57 Pythagoras theorem)=> AX = 38/17 003. BO = OY , BO = 2/1 (Ratio Chasing)=> OY = 2/1 004. AO = 3/1 , OY = 2/1 , AY ⊥ OY (Pythagoras Verification)=> Pythagorean Theorem's premises on Y, A, O are satisfied 005. Pythagorean Theorem's premises on Y, A, O are satisfied (r57 Pythagoras theorem)=> AY = 38/17 006. AX = 38/17 , AY = 38/17 (Ratio Chasing)=> AX = AY 007. BO = OX , BO = OY (Ratio Chasing)=> OX = OY 008. AX = AY , OX = OY (r23 Bisector is perpendicular)=> AO ⊥ XY It seems I've reached a dead end; Each followed by a three-digit number (e.g., 001, 002) represents a step in my reasoning process. Could you please tell me at which step I started to go wrong?

(1) Step 5;
**(2) Step 4;**
(3) Step 7;
(4) Step 6

(c) Faulty Branch Localization

Figure 7: Visualization Examples

Table 10: Geometric Rules (Part 1)

| Rule Name | Rule Statement |
| --- | --- |
| Definition of circle | Four points A, B, C, D equidistant from a center O all lie on a same circle. (One side of the definition of a circle.) |
| Arc determines internal angles | Two angles with the vertices P, Q on a circle that determine the same arc AB on that same circle are congruent. |
| Congruent angles are in a circle | If P, Q are vertices of congruent angles, and A and B are the intersections of the legs of the angles with vertices P and Q, there is a circle through A, B, P, and Q. |
| Same arc same chord | Arcs of the same length determine chords of the same length on the same circle. |
| Base of half triangle | The line connecting the midpoints of two sides of a triangle is parallel to the third side of the same triangle. |
| Thales Theorem I | Two parallel lines AB and CD cut by two intersecting transverse lines AC and BD, will determine a collection of proportional segments. The original statement of this rule did not have the non-degeneracy condition ncoll O A B as a hypothesis. |
| Bisector theorem I | If a line through a vertex of a triangle cuts the opposite side into two segments that are in proportion as the neighboring sides of the triangle, the line bisects the angle at the vertex it cuts. |
| Bisector theorem II | The internal bisector of a vertex of a triangle divides the opposite side into two segments that are in proportion to the neighboring sides of the triangle. |
| Isosceles triangle equal angles | The base angles of an isosceles triangle are congruent. |
| Equal base angles imply isosceles | If the base angles of a triangle are congruent, the triangle is isosceles. |
| Arc determines inscribed angles (tangent) | An inscribed angle determining that same arc will be congruent to the angle determining that arc with one leg being the tangent line at the vertex of the arc. |
| Same arc giving tangent | If two angles with vertices on a circle see the same arc, but one vertex is also an extremal point of the arc, a leg of the angle through this extremal point is the tangent to the circle at that same point. |
| Central angle vs inscribed angle I | The central angle doubles the inscribed angle when both determine the same arc in a circle. It mentions bisects the chord as an hypotheis instead of halves the angle because midpoint of a segment is a predicate, while bisector of an angle is not. |
| Central angle vs inscribed angle II | If a central angle has the same measure as a given inscribed angle on a circle, it will cut the chord corresponding to that angle in half. |
| Hypotenuse is diameter | The hypothenuse of a right triangle is a diameter of its circumcircle, or that the midpoint of the hypothenuse is the circumcenter of the right triangle. |
| Diameter is hypotenuse | If two points are the edges of the diameter of a circle, and at the same time are vertices of an inscribed triangle, the triangle has a right angle at the third vertex. |
| Cyclic trapezoid | A cyclic trapezoid is isosceles (refering specifically to the congruence of the angles on a base). |
| Bisector Construction | The perpendicular line through the midpoint of the segment is the perpendicular bisector of the segment (the locus of all equidistant points to the vertices of the segment). |
| Bisector is perpendicular | The locus of the points that are equidistant to the two vertices of a segment AB is a straight line perpendicular to AB. |
| Cyclic kite | A cyclic kite is always formed by two symmetric right triangles. |

Table 11: Geometric Rules (Part 2)

| Rule Name | Rule Statement |
| --- | --- |
| Diagonals of parallelogram I | If two segments intersect at their common midpoint, the vertices of the segments are the vertices of a parallelogram. |
| Diagonals of parallelogram II | The diagonals of a parallelogram meet at their common midpoint. |
| Thales theorem II | If two points C and D split to legs of a triangle on the same ratio, the line CD will be parallel to the base of the triangle. |
| Overlapping parallels | Two intersecting parallel lines are actually the same. |
| Midpoint is an eqratio | Midpoints split segments in the same ratio (1:2). |
| AA Similarity of triangles (Direct) | A similarity condition for a pair of triangles: that of two pairs of congruent angles between the triangles (angle-angle similarity). |
| AA Similarity of triangles (Reverse) | A similarity condition for a pair of triangles: that of two pairs of congruent angles between the triangles (angle-angle similarity). |
| Thales theorem IV | If three parallel lines are cut by two other lines, there is a corresponding pair of proportional segments determined by the intersection points. |
| Recognize center of cyclic (circle) | If three points lie on a circle with a known center, and there is a fourth point on that circle, the distance of the center of the circle to this fourth point is the same as to other points in a circle. |
| Midpoint splits in two | This rule converts a symbolic statement (M is the midpoint of AB) into an algebraic one (the ratio between AM and AB is 1:2). |
| Properties of similar triangles (Direct) | This rule goes from the pure statement that two triangles are similar to spilling out the corresponding statements about the proportion of the lengths of the sides and the equivalence of angles on both triangles. |
| Properties of similar triangles (Reverse) | This rule goes from the pure statement that two triangles are similar to spilling out the corresponding statements about the proportion of the lengths of the sides and the equivalence of angles on both triangles. |
| Definition of midpoint | This rule was created to detect midpoints by their defining axiomatic properties. It solidifies midp as a predicate. |
| Properties of midpoint (cong) | This rule extracts from the midp predicate the property that the midpoint is equidistant from the extremes of the segment. |
| Properties of midpoint (coll) | This rule extracts symbolically from the midp predicate the property that the midpoint is on the line of the segment. |
| Pythagoras theorem | If the proof state symbolically knows the three lengths of the sides of a right triangle ABC, and they satisfy that the sum of the squares of the lengths of the legs is equal to the square of the hypothenuse, it will add the proper orthogonal relation to the proof state. |
| Same chord same arc I | This rule gives conditions for inscribed angles on a circle defining chords of the same length to have the same measure. |
| Same chord same arc II | This rule gives conditions for inscribed angles on a circle defining chords of the same length to have the same measure. |
| SSS Similarity (Direct) | Proportional sides with same orientation imply similarity. |
| SSS Similarity (Reverse) | Proportional sides with opposite orientation imply similarity. |
| SAS Similarity (Direct) | Proportional sides with congruent included angle (same orientation) imply similarity. |
| SAS Similarity (Reverse) | Proportional sides with congruent included angle (opposite orientation) imply similarity. |
| Ratio Chasing | Manipulating and analyzing ratios between different quantities in a geometric setup to derive new relationships or prove geometric properties. |
| Numerical Check | Verifying geometric properties by numerical means, typically checking if numerical values of geometric entities satisfy expected relationships. |
| Angle Chasing | To pursue and derive angle relationships in geometric configurations, often using angle sum properties, parallel line angles, or circle angle properties. |
| Pythagoras Verification | The Pythagorean relationship ($a^2 + b^2 = c^2$) in right triangles, either confirming a triangle is right-angled or using a known right angle to derive side lengths. |

Table 12: Confidence-Intervals(%).

| | Level 1 | | Level 2 | | Level 3 | Level 4 |
|---|---|---|---|---|---|---|
| | **N.P.** | **S.P.** | **I.P.F.** | **S.D.** | **T.S.** | **F.B.L.** |
| Qwen2-VL-7b (Wang et al., 2024) | 55.16±4.26 | 19.51±2.13 | 26.17±1.90 | 36.84±2.59 | 25.25±3.39 | 19.96±4.88 |
| Qwen2-VL-72b (Wang et al., 2024) | 84.33±4.28 | 28.15±3.68 | 30.17±1.90 | 60.34±4.37 | 35.37±4.01 | 21.54±4.26 |
| Qwen2.5-VL-7b (Bai et al., 2025) | 86.90±3.93 | 20.00±3.18 | 30.84±2.59 | 40.67±2.59 | 35.76±2.90 | 24.49±2.93 |
| Qwen2.5-VL-72b (Bai et al., 2025) | 85.72±4.44 | 39.51±3.83 | 39.00±1.24 | 76.00±2.48 | 45.82±3.73 | 25.85±2.93 |
| GPT-4o (OpenAI, 2024b) | 68.65±4.26 | 22.71±1.06 | 59.50±4.97 | 43.00±2.48 | 37.27±4.73 | 23.36±0.98 |

Table 13: Confidence-Scores.

| | Level 1 | | Level 2 | | Level 3 | Level 4 |
|---|---|---|---|---|---|---|
| | **N.P.** | **S.P.** | **I.P.F.** | **S.D.** | **T.S.** | **F.B.L.** |
| Qwen2-VL-7b (Wang et al., 2024) | 0.97 | 0.96 | 0.97 | 0.97 | 0.95 | 0.90 |
| Qwen2-VL-72b (Wang et al., 2024) | 0.98 | 0.95 | 0.97 | 0.97 | 0.95 | 0.92 |
| Qwen2.5-VL-7b (Bai et al., 2025) | 0.98 | 0.94 | 0.97 | 0.97 | 0.97 | 0.95 |
| Qwen2.5-VL-72b (Bai et al., 2025) | 0.98 | 0.96 | 0.99 | 0.99 | 0.97 | 0.95 |
| GPT-4o (OpenAI, 2024b) | 0.97 | 0.98 | 0.97 | 0.98 | 0.95 | 0.98 |

# D DETAILS OF GEOBENCH EVALUATION SET

## D.1 VISUALIZATION EXAMPLE OF DIFFERENT TASK

In this section, as illustrated in Fig. 6 and Fig. 7, we provide a comprehensive description of the four levels and six exemplary instances of geometric problems, along with sample option descriptions, in greater detail.

## D.2 RULES USED

This section presents the complete set of geometric rules utilized by TrustGeoGen to construct reasoning graphs and those applied in GeoBench Task 5 (Theorem Selection). All rules are systematically categorized and listed in Table 10 and Table 11. These rules define the foundational logic for geometric reasoning within our benchmark framework.

# E CONFIDENCE ASSESSMENT

We have conducted additional replicate experiments and incorporated metrics such as confidence intervals and confidence scores, with the results presented in the Table 12 and 13. Among these, the Confidence-Score serves as a numerical indicator of data variability (with higher values corresponding to lower variability). As can be observed from the table, the testing results generally exhibit low variability, indicating a high level of reliability.

