# OpenReview forum: "GeoBench: Rethinking Multimodal Geometric Problem-Solving via Hierarchical Evaluation"
_ICLR.cc/2026/Conference — ICLR 2026 Poster_

### Official Review · Reviewer_YWp2 · 2025-10-26

**Soundness:** 2
**Presentation:** 2
**Contribution:** 3
**Rating:** 4
**Confidence:** 4

**Summary:**

The paper introduces GeoBench, a hierarchical benchmark for multimodal geometric problem solving. It defines four reasoning levels: Visual Perception, Goal-Oriented Planning, Rigorous Theorem Application, and Self-Reflective Backtracking, instantiated via six multiple-choice tasks built from problems generated and formally verified by TrustGeoGen. The dataset has 1021 items and is used to evaluate a range of general and reasoning MLLMs. Key empirical findings: (i) performance drops as task difficulty rises; (ii) sub-goal decomposition and irrelevant-premise filtering correlate most with final-answer accuracy; (iii) Chain-of-Thought can hurt faulty-branch localization; (iv) some OOD generalization is observed to GeoQA/Geometry3K.

**Strengths:**

- Hierarchical evaluation design. The four-level tasks with difficulty level increase is principled, and the task design reflects real-life use cases.

- Using TrustGeoGen with verified reasoning graphs reduces label noise.

- Comprehensive evaluations cover many popular general and reasoning MLLMs and CoT ablation highlighting task-specific limits.

**Weaknesses:**

* While TrustGeoGen with formal verification is a strength, the heavy reliance on synthetic problems may narrow the distribution. It remains unclear how faithfully the tasks reflect real contest or classroom diagrams.

* Results are reported mostly as point accuracies; significance tests, confidence intervals, or bootstrap estimates are missing.

* A random-choice baseline is trivial; a properly calibrated human baseline (e.g., novice vs. expert) is needed to contextualize difficulty and headroom.

* Inaccurate or overstated claims in results discussion.

  * The statement around lines 346–348 (e.g., *“... T.S. task where other models generally perform poorly.”*) is not accurate, as **o1** attains similar performance.
  * The claim *“Reasoning MLLMs consistently outperform General MLLMs … confirming their superior reasoning capabilities …”* overgeneralizes; several non-reasoning models outperform some reasoning models on certain tasks. These statements should be revised to reflect the actual per-task and per-model variability.

* Some benchmark details require further clarification (see the Questions section).

**Questions:**

* What is difference between GeoBench and GeoBench-Solving, is the later a subset of the former one?

* The results in Table 4 and Table 5 have a lot overlap, which may not necessary.

* A proper reference for TrustGeoGen at line 155 is missing.

* Important evaluation details such as temperature, max tokens, retry logic, and related decoding parameters are not fully reported.

* In Section 3.2, how the distractors are designed?

* In Table 4, the DeepSeek-R1 results is obtained via image textual descriptions. This may cause unfair comparison as models may benefit from accurate image textual descriptions, especially for geometric problem solving. How the other models perform when the image is replaced by the textual descriptions?

* Some results in Table 4 seems unintuitive. For example, Qwen2.5-VL-7b performs pretty well on N.P. tasks but suddenly very poor on S.P. tasks, which all belong to Level 1 tasks.

*  In Section 4.3, how the feature vectors are generated? I am a bit confused by this section.

*  In Section 4.4, why the other two benchmarks are OOD? What is the point of studying other two benchmarks in your paper?

*  In Section 4.5, it would help to include side-by-side qualitative examples with vs. without CoT, highlighting where CoT helps or hurts. Annotated failure cases would strengthen the argument.

---

> ### Author Response · Authors · 2025-11-25
>
> ### Weaknesses:
>
> > **Q1. While TrustGeoGen with formal verification is a strength, the heavy reliance on synthetic problems may narrow the distribution. It remains unclear how faithfully the tasks reflect real contest or classroom diagrams.**
>
> Thank you for your valuable suggestions. The t-SNE visualization in Figure 2 and the length distribution of solution tokens in Figure 2(c) compare the data distributions between GeoBench and the **GeoQA dataset (a real-world scene dataset)**. These figures demonstrate that our benchmark exhibits **a broader distribution compared to GeoQA**. In Section 3.4, we further compare the difficulty levels of our benchmark with **three real-world scene datasets**—GeoQA, Geometry3K, and OlympiadBench-Geo—and conclude that the difficulty of our dataset exceeds that of typical middle school mathematics problems and **approaches the level of Olympiad-style questions**.
>
> Regarding **real-world scene datasets**, they currently fall outside the scope of our framework and cannot be formally structured into specific tasks. We attempted to employ existing tools for style rendering (converting images into hand-drawn styles), but found that alphanumeric annotations on geometric images, being relatively small, were consistently overlooked by these tools. Consequently, we will continue to explore appropriate methodologies and style rendering techniques, with the objective of successfully transforming our images into realistic scene representations for inclusion in the dataset and subsequent evaluation.
>
> > **Q2. Results are reported mostly as point accuracies; significance tests, confidence intervals, or bootstrap estimates are missing.**
>
> We thank the reviewer for their valuable suggestion. We have conducted additional replicate experiments and incorporated metrics such as confidence intervals and confidence scores, with the results presented in the table below:
>
> | Model          | NP           | SP           | IPF          | SD           | TS           | FBL          |
> |----------------|--------------|--------------|--------------|--------------|--------------|--------------|
> | Qwen2-VL-7B    | 55.16±4.26%  | 19.51±2.13%  | 26.17±1.90%  | 36.84±2.59%  | 25.25±3.39%  | 19.96±4.88%  |
> | Qwen2-VL-72B   | 84.33±4.28%  | 28.15±3.68%  | 30.17±1.90%  | 60.34±4.37%  | 35.37±4.01%  | 21.54±4.26%  |
> | Qwen2.5-VL-7B  | 86.90±3.93%  | 20.00±3.18%  | 30.84±2.59%  | 40.67±2.59%  | 35.76±2.90%  | 24.49±2.93%  |
> | Qwen2.5-VL-72B | 85.72±4.44%  | 39.51±3.83%  | 39.00±1.24%  | 76.00±2.48%  | 45.82±3.73%  | 25.85±2.93%  |
> | GPT-4o         | 68.65±4.26%  | 22.71±1.06%  | 59.50±4.97%  | 43.00±2.48%  | 37.27±4.73%  | 23.36±0.98%  |
>
> |Confidence-Score|NP|SP|IPF|SD|TS|FBL|
> |-|-|-|-|-|-|-|
> |Qwen2-VL-7B|0.97|0.96|0.97|0.97|0.95|0.90|
> |Qwen2-VL-72B|0.98|0.95|0.97|0.97|0.95|0.92|
> |Qwen2.5-VL-7B|0.98|0.94|0.97|0.97|0.97|0.95|
> |Qwen2.5-VL-72B|0.98|0.96|0.99|0.99|0.97|0.95|
> |GPT-4o|0.97|0.98|0.97|0.98|0.95|0.98|
>
> Among these, the Confidence-Score serves as a numerical indicator of data variability (with higher values corresponding to lower variability). As can be observed from the table, the testing results generally exhibit low variability, indicating a high level of reliability.
>
> ***We have added confidence assessments to the revised version of the article. Please refer to Section 4.3 and Appendix E.***
>
> > **Q3. A random-choice baseline is trivial; a properly calibrated human baseline (e.g., novice vs. expert) is needed to contextualize difficulty and headroom.**
>
> We equally distributed the data to five human experts (each holding a Ph.D. from a top 1% national university) for manual evaluation. The results, calculated as the arithmetic mean with the exclusion of the highest and lowest ratings, are as follows:
>
> ||NP|SP|IPF|SD|TS|FBL|
> |-|-|-|-|-|-|-|
> |Human Results|100.00%|100.00%|77.78%|100.00%|56.67%|52.94%|
> |MLLMs' Best Results|87.50%|74.81%|74.00%|91.00%|54.39%|27.89%|
>
> As shown in table, human experts consistently achieved higher accuracy than MLLMs across all tasks. This performance gap was particularly pronounced for the **NP** and **SP** tasks, which require image-grounding, and the **FBL** task, which involves more complex reasoning.
>
> ***We have added human results and corresponding analysis to the revised version of the article. Please refer to Section 4.2.***

---

> ### Author Response · Authors · 2025-11-25
>
> > **Q4. Inaccurate or overstated claims in results discussion.**
> > 1) The statement around lines 346–348 (e.g., “... T.S. task where other models generally perform poorly.”) is not accurate, as o1 attains similar performance.
> > 2) The claim “Reasoning MLLMs consistently outperform General MLLMs … confirming their superior reasoning capabilities …” overgeneralizes; several non-reasoning models outperform some reasoning models on certain tasks. These statements should be revised to reflect the actual per-task and per-model variability.
>
> We appreciate the reviewer for pointing out the issues with expression. ***We sincerely apologize for the inaccuracies in our wording and have revised these accordingly in the updated revision. Please refer to Section 4.3.***
>
> 1) "Proprietary commercial models demonstrate superior reasoning capabilities. For instance, OpenAI-o3 (OpenAI, 2025) achieves 91.00% accuracy on the S.D. task and 54.39% accuracy on the T.S. task, while OpenAI-o1 also attains a comparable accuracy of 53.22% on the T.S. task—significantly outperforming most other models which generally exhibit poor performance on these tasks."
>
> 2) "Across most tasks, reasoning MLLMs outperform general MLLMs. Top-performing models such as OpenAI-o1, OpenAI-o3, Claude-3.7-sonnet, and Gemini-2.5-pro surpass 70% of general models in N.P. task, exceed all general models in S.P. and I.P.F. tasks, and outperform over 90% of general models in S.D. and T.S. tasks."
>
> ### Questions:
>
> > **Q1. What is difference between GeoBench and GeoBench-Solving, is the later a subset of the former one?**
>
> **GeoBench-solving** focuses specifically on geometric problems where the labels correspond directly to the final answers. This dataset evaluates solely based on answer correctness, considering a response accurate if the final answer is correct, without examining the underlying reasoning process. In contrast, **GeoBench** is designed to assess the model's reasoning process through six distinct tasks. Each task is structured to evaluate whether the model possesses specific capabilities, such as numerical perception and subtask decomposition. This benchmark enables a more thorough and nuanced evaluation of models' reasoning abilities.
>
> > **Q2. The results in Table 4 and Table 5 have a lot overlap, which may not necessary.**
>
> Thank you for raising this point. While there is indeed partial data overlap between Tables 4 and 5, Table 5 specifically emphasizes the comparative evaluation across the GeoBench-Solving, GeoQA, and Geometry3K benchmarks, incorporating three additional columns of data. Integrating these into Table 4 would result in excessive data density and compromise visual clarity, which motivated our decision to present them separately.
>
> > **Q3. A proper reference for TrustGeoGen at line 155 is missing.**
>
> We sincerely appreciate your thorough review and valuable feedback. ***We have addressed these points in the revised version of the manuscript. Please refer to Line 154.***
>
> > **Q4. Important evaluation details such as temperature, max tokens, retry logic, and related decoding parameters are not fully reported.**
>
> For all MLLMs, we set the parameters as `temperature = 0.95`, `max_tokens = 8192`, and `top_p = 0.7`. For API-based models (including OpenAI-o1/o3, GPT-4o/4v, DeepSeek-R1, Claude-3.7-sonnet, and Gemini-2.5-pro), we implemented a `retry logic` that allows up to 5 API reattempts in case of network connection errors.
>
> ***We have added parameter configuration to the revised version of the article. Please refer to Section 4.1 and Appendix C.1.***
>
> > **Q5. In Section 3.2, how the distractors are designed?**
>
> The design principle for distractors is to avoid overlap with the correct option while ensuring randomness. The primary steps for generating correct and incorrect options across the six tasks are as follows:
>
> In the **N.P. task**, one correct answer is randomly selected from the numerical premises of the problem, and three incorrect options are generated through numerical modifications and label alterations. In the **S.P. task**, a correct answer is randomly chosen from the geometric relational premises, and three incorrect options are generated by applying inverse negation to other relations. For the **I.P.F. task**, one correct option is selected from all unused premises, while one incorrect option is chosen from the premises that have been used. In the **S.D. task**, the correct option is defined as the set of predecessor states of the final state excluding the initial conditions. The incorrect options consist of state subsets that do not correspond to the correct set. The **T.S. task** uses an unused theorem as the correct option, with three randomly selected applied theorems serving as incorrect options. In the **F.B.L. task**, an erroneous reasoning chain is provided, where the step number containing the error is designated as the correct option, and three other randomly chosen step numbers form the incorrect options.

---

> ### Author Response · Authors · 2025-11-25
>
> > **Q5. In Section 3.2, how the distractors are designed?**
>
> ***A more detailed description of task generation has been provided in Section 3.2 of the revised version of the paper.***
>
> > **Q6. In Table 4, the DeepSeek-R1 results is obtained via image textual descriptions. This may cause unfair comparison as models may benefit from accurate image textual descriptions, especially for geometric problem solving. How the other models perform when the image is replaced by the textual descriptions?**
>
> We appreciate the reviewer's inquiry. Firstly, all multimodal large language models in our evaluation received both textual and visual inputs, with the exception of DeepSeek-R1 where visual inputs were omitted due to its architectural constraints—though its textual inputs remained consistent with other models. Secondly, we conducted text-only ablation experiments on MLLMs, with results as follows:
>
> ||NP|SP|IPF|SD|TS|FBL|
> |-|-|-|-|-|-|-|
> |Qwen2-VL-7B with text|33.93%|23.70%|29.50%|36.00%|23.71%|22.45%|
> |Qwen2-VL-7B with image+text|54.17%|18.52%|27.00%|36.00%|23.71%|17.69%|
> |Qwen2-VL-72B with text|47.62%|19.26%|27.50%|59.50%|30.99%|21.09%|
> |Qwen2-VL-72B with image+text|86.31%|29.63%|31.00%|60.50%|37.11%|21.77%|
> |Qwen2.5-VL-7B with text|39.29%|19.26%|28.00%|40.00%|28.65%|24.49%|
> |Qwen2.5-VL-7B with image+text|87.50%|18.52%|32.00%|41.00%|37.11%|25.17%|
> |Qwen2.5-VL-72B with text|51.19%|22.96%|39.00%|75.00%|43.27%|21.77%|
> |Qwen2.5-VL-72B with image+text|85.71%|40.74%|38.50%|77.00%|47.42%|26.53%|
>
> Experimental findings indicate that most vision-language models exhibit inferior evaluation performance when processing text-only inputs compared to multimodal inputs incorporating both images and text. We hope this clarification addresses the reviewer's concerns.
>
> ***We have added text-only experiments to the revised version of the article. Please refer to Section 4.8.***
>
> > **Q7. Some results in Table 4 seems unintuitive. For example, Qwen2.5-VL-7b performs pretty well on N.P. tasks but suddenly very poor on S.P. tasks, which all belong to Level 1 tasks.**
>
> The **N.P.** task bears some resemblance to OCR, as models demonstrate relatively strong performance by primarily recognizing basic elements such as numbers and letters in the images. In contrast, the **S.P.** task requires a deeper understanding of geometric figures, presenting greater difficulty—particularly with certain complex shapes in the dataset. For instance: [Fig.1](https://anonymous.4open.science/api/repo/ICLR26_Figures-C218/file/143.png?v=7d86d22c) [Fig.2](https://anonymous.4open.science/api/repo/ICLR26_Figures-C218/file/2648.png?v=4782a046).
>
> > **Q8. In Section 4.3, how the feature vectors are generated? I am a bit confused by this section.**
>
> In our benchmark, for a given task (e.g., N.P.), the accuracy results of all MLLMs on that task are treated as a feature vector (i.e., a column in the table, such as $[28.57\%, 25.00\%, ..., 87.50\%]$). The same approach is applied to the GeoBench-solving, GeoQA, and Geometry3K datasets. The Spearman correlation coefficient between two feature vectors is used to represent the direction and strength of their relationship.
>
> > **Q9. In Section 4.4, why the other two benchmarks are OOD? What is the point of studying other two benchmarks in your paper?**
>
> GeoQA and Geometry3K comprise **real-world data**, which differ from our dataset in both generation methodology and distribution.
>
> In Section 4.4, we investigate the correlation between the six tasks in our benchmark and final answer performance on the GeoBench-Solving dataset. To validate whether the relationship between model capabilities on these six tasks and final answer performance generalizes to other datasets, we extended our analysis to GeoQA and Geometry3K. The observed results demonstrate consistent similarity across datasets, thereby **verifying the effectiveness** of our evaluation framework.
>
> > **Q10. In Section 4.5, it would help to include side-by-side qualitative examples with vs. without CoT, highlighting where CoT helps or hurts. Annotated failure cases would strengthen the argument.**
>
> We greatly appreciate your suggestion to analyse comparative cases of Chain-of-Thought (CoT) enabled/disabled, to clearly demonstrate the boundaries of CoT's capability in geometric reasoning: when it is effective and when it fails.
>
> 1) Case Where **CoT Helps**
>
>      In Task3, CoT assists models in explicitly ruling out irrelevant geometric premises. For example, when determining whether a ratio such as AD/DF=CE/EG can be derived, the CoT-enabled model correctly reasoned that the relation follows solely from construction equalities. It therefore rejected distractors containing unnecessary angle or perpendicular constraints. By contrast, the no-CoT model selected an option with redundant premises, failing to articulate why those conditions were irrelevant.

---

> ### Author Response · Authors · 2025-11-25
>
> > **Q10. In Section 4.5, it would help to include side-by-side qualitative examples with vs. without CoT, highlighting where CoT helps or hurts. Annotated failure cases would strengthen the argument.**
>
> 2) Case Where **CoT Hurts**
>
>    Task 5 is an inverse exclusion task that requires identifying the rule not used in a detailed proof trace. In this task, the CoT mode sometimes introduced a flawed reasoning chain. While attempting to trace the usage of all rules, the generated steps led to a deviation, resulting in the selection of an incorrect option. In Task 6, the prompt includes a reasoning chain that contains an intentionally incorrect step. CoT always reduces accuracy here: by expanding the provided chain, the model often reproduces the flawed logic and points to a later step as the first error. It illustrates that how CoT's generated reasoning process can be misleading in complex rule discrimination scenarios.

---

> ### Comment · Reviewer_YWp2 · 2025-11-25
>
> Thanks for your responses. Regarding the text-only input to MLLMs, I was thinking you will replace image caption with the actual image to the model. Such study will decouple the visual perception failures from reasoning failures. However, it seems that for some tasks, without images gives even better performance, meaning that the visual input may not that necessary to solve the task. Some insights on this finding will be great. Otherwise, I have no other questions. I will increase my score in light of authors detailed responses.

---

> > ### Author Response · Authors · 2025-11-26
> >
> > We are grateful for your kind words about our research and we sincerely appreciate your insightful comments!

---

### Official Review · Reviewer_sFVL · 2025-10-29

**Soundness:** 3
**Presentation:** 4
**Contribution:** 3
**Rating:** 8
**Confidence:** 4

**Summary:**

The paper proposes GeoBench, a hierarchical benchmark to evaluate geometric reasoning in multimodal large language models. It introduces four levels: visual perception, goal-oriented planning, rigorous theorem application, and self-reflective backtracking which is implemented through six tasks derived from formally verified problems generated using TrustGeoGen. Across 1,021 examples, the authors show that reasoning-oriented models like OpenAI-o3 outperform general MLLMs, though performance decreases sharply with task complexity. The analysis highlights that sub-goal decomposition and irrelevant premise filtering are key determinants of geometric problem-solving accuracy and that Chain-of-Thought prompting is not universally beneficial.

**Strengths:**

I liked the clarity in the paper's writing and the results are comprehensive, and have two strengths to highlight:

- Hierarchical evaluation grounded in cognitive theory: The benchmark’s structure, inspired by the van Hiele model, allows precise diagnosis of reasoning abilities rather than measuring final-answer accuracy alone.

- Comprehensive and formally verified dataset: Using TrustGeoGen ensures rigorous, contamination-free problem generation, making GeoBench a strong diagnostic tool for evaluating how MLLMs reason through geometric logic.

**Weaknesses:**

The benchmark relies on synthetic, clean diagrams and controlled premises. This limits assessment of robustness to real-world variability such as hand-drawn figures, scanned textbook noise, ambiguous markings, and imperfect annotations. Adding a real-diagram slice or perturbation suite would strengthen ecological validity.

**Questions:**

I do not have any particular concerns with the paper, except maybe including some more MLLMs such as Molmo, Math-LLaVA and MathPuma. The authors can show some preliminary results on these models

---

> ### Author Response · Authors · 2025-11-25
>
> ### Weaknesses:
>
> > **Q. The benchmark relies on synthetic, clean diagrams and controlled premises. This limits assessment of robustness to real-world variability such as hand-drawn figures, scanned textbook noise, ambiguous markings, and imperfect annotations. Adding a real-diagram slice or perturbation suite would strengthen ecological validity.**
>
> We sincerely thank you for your valuable suggestion. We first applied **random rotations of $\pm20°$** to the images to create a perturbed test set and conducted corresponding experiments. The results are presented in the table below.
>
> ||NP|SP|IPF|SD|TS|FBL|
> |-|-|-|-|-|-|-|
> |Qwen2-VL-7B|57.14%|23.70%|25.50%|38.00%|26.32%|22.45%|
> |Qwen2-VL-72B|80.95%|33.34%|28.00%|56.50%|39.77%|19.73%|
> |Qwen2.5-VL-7B|79.17%|20.74%|28.50%|35.50%|35.09%|23.13%|
> |Qwen2.5-VL-72B|83.93%|40.00%|39.00%|76.00%|44.44%|24.49%|
>
> The evaluation results for the **original image** are presented in the table below:
> ||NP|SP|IPF|SD|TS|FBL|
> |-|-|-|-|-|-|-|
> |Qwen2-VL-7B|54.17%|18.52%|27.00%|36.00%|23.71%|17.69%|
> |Qwen2-VL-72B|86.31%|29.63%|31.00%|60.50%|37.11%|21.77%|
> |Qwen2.5-VL-7B|87.50%|18.52%|32.00%|41.00%|37.11%|25.17%|
> |Qwen2.5-VL-72B|85.71%|40.74%|38.50%|77.00%|47.42%|26.53%|
>
> Based on a comparative analysis with the evaluation results of the **Qwen MLLMs** presented in Table 4 of the paper, we observe that the performance metrics remain remarkably consistent. This finding empirically demonstrates that minor image perturbations (within $\pm20°$ rotation) do not significantly affect the evaluation outcomes.
>
> ***We have added the results and corresponding analysis of the perturbation test set to the revised version of the article. Please refer to Section 4.7.***
>
> Regarding **real-world scene datasets**, they currently fall outside the scope of our framework and cannot be formally structured into specific tasks. We attempted to employ existing tools for style rendering (converting images into hand-drawn styles), but found that alphanumeric annotations on geometric images, being relatively small, were consistently overlooked by these tools. Consequently, we will continue to explore appropriate methodologies and style rendering techniques, with the objective of successfully transforming our images into realistic scene representations for inclusion in the dataset and subsequent evaluation.
>
> ### Questions:
> > **Q. I do not have any particular concerns with the paper, except maybe including some more MLLMs such as Molmo, Math-LLaVA and MathPuma. The authors can show some preliminary results on these models.**
>
> We sincerely thank the reviewer for your valuable suggestion. We have preliminarily deployed the **Molmo[1]** model (an open-source vision-language model trained on the PixMo[1] dataset, a collection of image-text pairs) and conducted experiments, with the results presented in the table below:
> ||NP|SP|IPF|SD|TS|FBL|
> |-|-|-|-|-|-|-|
> |Molmo-72B|78.57%|12.59%|32.00%|31.00%|26.90%|18.36%|
>
> Based on the results presented in the table, it can be observed that the performance of the Molmo model falls short of that achieved by proprietary commercial MLLMs models, indicating its relatively limited capability in geometric reasoning.
>
> ***We have incorporated Molmo's experimental results and corresponding analysis into the revised version of the article. Please refer to Table 4 and Section 4.3.***
>
> ---
> ***Reference***
>
> [1] Deitke, Matt, et al. "Molmo and pixmo: Open weights and open data for state-of-the-art multimodal models." arXiv e-prints (2024).

---

> > ### Comment · Reviewer_sFVL · 2025-11-27
> >
> > Thank you for addressing my concerns. I appreciate the authors having experimented on the perturbation effects, and hope that they can extend their work in the future to more real-world scenarios which involve other ambiguities. Also, the experiments on Molmo are helpful and do indeed provide clarity on the performative differences between propietary MLLMs and open-source models. I do not have any additional clarifications required from my end and will be standing by my score.

---

> > > ### Author Response · Authors · 2025-11-27
> > >
> > > Thank you! We sincerely appreciate your positive assessment of our work.

---

> ### Author Response · Authors · 2025-11-27
>
> Dear Reviewer sFVL,
>
> We sincerely appreciate your valuable feedback on the paper. Given that the deadline for the discussion phase is approaching, we would be most grateful if you could kindly share any remaining concerns at your earliest convenience, and we will address them immediately.

---

### Official Review · Reviewer_aj3h · 2025-10-30

**Soundness:** 2
**Presentation:** 3
**Contribution:** 3
**Rating:** 4
**Confidence:** 4

**Summary:**

Addressing three major limitations in current geometric reasoning evaluation—the risk of test data contamination, overemphasis on final answers, and insufficient diagnostic granularity—this paper proposes a more scientific and refined hierarchical benchmark called GeoBench. Through systematic evaluation, this benchmark not only reveals the capability limitations of existing AI models in handling complex geometric problems but, more importantly, precisely diagnoses the core factors affecting model performance.

**Strengths:**

1. It identifies key limitations in geometric reasoning evaluation (e.g., data contamination, overemphasis on answers) and addresses them through GeoBench—a hierarchical benchmark that decomposes geometric reasoning into distinct stages

2. The benchmark leverages the TrustGeoGen methodology to generate tasks verified for logical rigor, ensuring data novelty and mitigating contamination risks. This establishes a reliable foundation for equitable model evaluation.

**Weaknesses:**

--The evaluation framework lacks a necessary human verification step. Given the complexity of the dataset problems (as shown in Figure 4), establishing a performance baseline from human experts is crucial. Furthermore, the scope of evaluation should be expanded to include advanced mathematical reasoning agents—particularly those capable of using tools for exploration or constructing auxiliary lines—in order to assess the true capabilities of current models under problem-solving paradigms that closely resemble human approaches.

--Regarding the reasoning graphs relied upon in the synthetic data generation process, key details—such as the quality of these graphs and the fidelity with which models adhere to them—are not sufficiently elaborated. This raises concerns about potential risks to synthetic data quality: if models do not strictly follow the intended reasoning logic, or if problems are constructed based on flawed reasoning graphs, the validity and reliability of the resulting problems may be called into question.

**Questions:**

N/A

---

> ### Author Response · Authors · 2025-11-25
>
> ### Weakness
>
> > **Q1. The evaluation framework lacks a necessary human verification step.**
>
> We equally distributed the data to five human experts (each holding a Ph.D. from a top 1% national university) for manual evaluation. The results, calculated as the arithmetic mean with the exclusion of the highest and lowest ratings, are as follows:
>
> ||NP|SP|IPF|SD|TS|FBL|
> |-|-|-|-|-|-|-|
> |Human Results|100.00%|100.00%|77.78%|100.00%|56.67%|52.94%|
> |MLLMs' Best Results|87.50%|74.81%|74.00%|91.00%|54.39%|27.89%|
>
> As shown in table, human experts consistently achieved higher accuracy than MLLMs across all tasks. This performance gap was particularly pronounced for the **NP** and **SP** tasks, which require image-grounding, and the **FBL** task, which involves more complex reasoning.
>
> ***We have added human results and corresponding analysis to the revised version of the article. Please refer to Section 4.2.***
>
> > **Q2. The scope of evaluation should be expanded to include advanced mathematical reasoning agents—particularly those capable of using tools for exploration or constructing auxiliary lines.**
>
> Regarding advanced mathematical reasoning agents capable of utilizing auxiliary lines, we conducted a preliminary investigation and identified **VisualSketch[1]** as an agent meeting the requirements. However, since its input is based on formal language while our data input is in natural language, the two are incompatible, preventing us from utilizing it for evaluation. Other agents, such as **Vista[2]**, are generally designed as general-purpose multimodal agents and are considerably distant from the context of geometric problems, making them unsuitable for evaluation in this benchmark.
>
> Finally, we also selected an agent called **CodePlot[3]** for testing, which can use Python to generate plots during the reasoning process to assist in thinking. The experimental results in table show that its performance is somewhat inferior to that of commercial large models, particularly with a significant disadvantage in T.S. and F.B.L. tasks.
>
> ||NP|SP|IPF|SD|TS|FBL|
> |-|-|-|-|-|-|-|
> |CodePlot|73.81%| 38.52%| 30.50%| 41.50% | 19.30%| 15.65% |
>
> We speculate that this may be because CodePlot overfitted its training data toward final answers, consequently lacking robust geometric reasoning capabilities, which further demonstrates the advantage of our hierarchical evaluation framework.
>
> ***We have incorporated CodePlot's experimental results and corresponding analysis into the revised version of the article. Please refer to Table 4 and Section 4.3.***
>
> > **Q3. Regarding the reasoning graphs relied upon in the synthetic data generation process, key details—such as the quality of these graphs and the fidelity with which models adhere to them—are not sufficiently elaborated.**
>
> The reasoning graph underlying synthetic data generation is produced by the **TrustGeoGen** data engine, which operates entirely on a rule-based approach without incorporating large language models. Each step employs correct rules to generate the correct subsequent state node, as elaborated in Section 3.1 of the paper.
>
> Furthermore, all six tasks in GeoBench are constructed based on this reasoning graph. Specifically: In the **N.P. task**, one correct answer is randomly selected from the numerical premises of the problem, and three incorrect options are generated through numerical modifications and label alterations. In the **S.P. task**, a correct answer is randomly chosen from the geometric relational premises, and three incorrect options are generated by applying inverse negation to other relations. For the **I.P.F. task**, one correct option is selected from all unused premises, while one incorrect option is chosen from the premises that have been used. In the **S.D. task**, the correct option is defined as the set of predecessor states of the final state excluding the initial conditions. The incorrect options consist of state subsets that do not correspond to the correct set. The **T.S. task** uses an unused theorem as the correct option, with three randomly selected applied theorems serving as incorrect options. In the **F.B.L. task**, an erroneous reasoning chain is provided, where the step number containing the error is designated as the correct option, and three other randomly chosen step numbers form the incorrect options.
>
> ***A more detailed description of task generation has been provided in Section 3.2 of the revised version of the paper.***
>
> ---
> ***Reference***
>
> [1] Hu, Yushi, et al. "Visual sketchpad: Sketching as a visual chain of thought for multimodal language models." Advances in Neural Information Processing Systems 37 (2024).
>
> [2] Huang, Zeyi, et al. "Visualtoolagent (vista): A reinforcement learning framework for visual tool selection." arXiv preprint (2025).
>
> [3] Duan, Chengqi, et al. "CodePlot-CoT: Mathematical Visual Reasoning by Thinking with Code-Driven Images." arXiv preprint (2025).

---

> ### Author Response · Authors · 2025-11-27
>
> Dear Reviewer aj3h,
>
> We sincerely appreciate your valuable feedback on the paper. Given that the deadline for the discussion phase is approaching, we would be most grateful if you could kindly share any remaining concerns at your earliest convenience, and we will address them immediately.

---

> > ### Comment · Reviewer_aj3h · 2025-11-28
> > **response**
> >
> > Thanks for your responses. Your response has addressed my concerns. The detailed discussion regarding some agent frameworks can be included in the next version. I will raise my score to "Positive".

---

### Official Review · Reviewer_FDHN · 2025-11-01

**Soundness:** 2
**Presentation:** 3
**Contribution:** 3
**Rating:** 6
**Confidence:** 3

**Summary:**

This work presents a benchmark to evaluate multimodal geometric reasoning by VLMs/MLLMs. The hierarchical evaluation proposed offers a novel assessment framework that goes beyond answer accuracy metrics. Additionally, this helps identify the exact steps at which models fail in the solving process, providing actionable insights into improving model performance. Finally, this work identifies the failure of the common CoT approach when it comes to geometric reasoning.

**Strengths:**

1. The hierarchical framework is a useful diagnostic tool to provide actionable insights into pitfalls in the reasoning process.
2. The benchmark presented in this work goes beyond data collation and adds a unique set of information for analysing model performance.

**Weaknesses:**

1, This work does not detail how the automatically generated tasks are verified for accuracy and legitimacy
2. Likewise, there is no insight into how the automatically-generated problems are distributed in terms of logical and reasoning complexity. In addition to the empirical comparison against established benchmarks and their levels, the work could benefit from a deeper, and qualitative, analysis of the complexity and difficulty of the problems in this benchmark,

**Questions:**

Questions:
1. How does this benchmark ensure diversity in the questions generated?
2. Can this generation and evaluation framework be extended to geometric problems in the 3D space?

Suggestions:
1. This work would benefit from further proofreading. Some errors that could be fixed:
- 1.1 L161: TrustGenGen -> TrustGeoGen
- 1.2 L250: Capitalizing the first letter of the section.

---

> ### Author Response · Authors · 2025-11-25
>
> ### Weakness:
> > **Q1. This work does not detail how the automatically generated tasks are verified for accuracy and legitimacy**
>
> Thank you for your suggestion. The six tasks in GeoBench are all constructed based on the inference graphs from TrustGeoGen, where the correctness of each step in these graphs is rigorously guaranteed (as detailed in Section 3.1 of the paper). Specifically, for the **N.P. task**, one correct answer is randomly selected from the numerical premises of the problem, and three incorrect options are generated through numerical modification and label modification. For the **S.P. task**, one correct answer is randomly chosen from the geometric relational premises, and three incorrect options are constructed by applying inverse negation to other relations.
>
> In the **I.P.F. task**, the correct option is selected from all unused premises, while the incorrect options are chosen from the set of premises actually used. The **S.D. task** defines the correct option as the set of preceding states of the final state excluding the initial conditions, with incorrect options composed of state subsets that do not belong to the correct set.
>
> For the **T.S. task**, an unused theorem serves as the correct option, and three randomly selected used theorems form the incorrect options. Lastly, the **F.B.L. task** provides an incorrect reasoning chain, where the step number containing the error is the correct option, and three other randomly selected steps constitute the incorrect options.
>
> ***More detailed task generation procedures have been elaborated in Section 3.2 of the revised manuscript.***
>
> > **Q2. there is no insight into how the automatically-generated problems are distributed in terms of logical and reasoning complexity.**
>
> Thank you for your suggestion.
>
> From the perspective of question generation, the questions produced by the **TrustGeoGen** data engine are constructed by iteratively adding geometric elements to a **base scenario**, forming formalized problems with corresponding premises. New geometric statements are then derived through the stepwise application of reasoning rules. For the base scenarios, we enhance basic geometric configurations (e.g., triangles, circles, rectangles) with augmented reality (AR) information (such as lengths and angles) and supplementary points (midpoints, perpendiculars, etc.) to establish geometric relationships. Through manual engineering, we have developed **40 base scenario functions**, covering most real-world geometric problem scenarios.
>
> Regarding the generation outcomes, in Section 3.3 of the paper, we plotted the distributions of images, solutions, and solution lengths, and compared them with the distributions in GeoQA. It can be observed that GeoBench exhibits relatively broad distributions, particularly in the distribution of solution lengths, which indicates the higher difficulty and complexity of our problem reasoning. Additionally, Section 3.4 of the paper discusses the difficulty analysis of GeoBench problems, suggesting that they are generally more challenging than standard middle school math problems and approach the difficulty level of Olympiad-style questions.
>
>
> ### Question:
>
> > **Q1:How does this benchmark ensure diversity in the questions generated?**
>
> Similar to the second point in Weakness, both our question generation process (involving multiple geometric elements and base scenes) and the results (comparison of data distribution, analysis of data difficulty) demonstrate the diversity of the generated questions.
>
> > **Q2. Can this generation and evaluation framework be extended to geometric problems in the 3D space?**
>
> This framework can be extended to geometric problems in three-dimensional space. By incorporating three-dimensional geometric elements, definitions, and rules into the data engine (TrustGeoGen), it becomes capable of generating solid geometry problems with fully correct reasoning steps. The evaluation criteria can consistently adopt our hierarchical framework.
>
> ### Suggestions:
>
> > **Q. Some errors that could be fixed.**
>
> We sincerely appreciate your careful review in identifying the errors in our manuscript.
>
> ***We have addressed this issue in the revised version of the article. Please refer to Line 161 and Line 312.***

---

> ### Author Response · Authors · 2025-11-27
>
> Dear Reviewer FDHN,
>
> We sincerely appreciate your valuable feedback on the paper. Given that the deadline for the discussion phase is approaching, we would be most grateful if you could kindly share any remaining concerns at your earliest convenience, and we will address them immediately.

---

### Author Response · Authors · 2025-11-30
**Summary of Rebuttal during ICLR Incident (Part 1 of 2)**

Dear Area Chairs,

In light of this unforeseen incident, we fully recognize the challenges confronting the ICLR organizing committee and the additional responsibilities you now shoulder. To facilitate your workflow, we have synthesized the principal contributions of our work, key rebuttal arguments, and score trajectories. A detailed exposition follows.

## Summary of Our Paper

This paper introduces GeoBench, a hierarchical evaluation benchmark designed to assess the geometric reasoning capabilities of multimodal large language models. The principal contributions and insights are summarized as follows:

- A four-tier progressive framework is established to systematically evaluate geometric reasoning, enabling fine-grained diagnosis of model capabilities beyond binary final-answer-based assessments.
- Correlation analyses reveal strong alignment between the six tasks in this benchmark and both Geo-Solving tasks and real-world datasets, supporting the ecological validity of the proposed benchmark.
- Chain-of-thought prompting demonstrates ineffectiveness in faulty branch localization tasks, suggesting that misleading reasoning steps in prompts may interfere with models’ ability to perform accurate error correction.
- We have more insights based on our evaluation framework, including that while multimodal inputs generally enhance model performance over text-only inputs, exceptions exist in minor cases. The benchmark also demonstrates robustness, as model performance remains stable under slight image rotations.

## Response to Reviewers' Comments

### FDHN (initial score: 6)
|Questions|Response|Addressed or Not?|
|-|-|-|
|How to guarantee the accuracy and rationale of the automatically generated tasks in this paper?|We provided a detailed explanation of the question generation process and incorporated this clarification into the revised version of the manuscript.|non-responsive|
|How the complexity and diversity of questions generated by GeoBench are ensured?|We provided a comprehensive elaboration on the process of question generation, which involves diverse geometric elements and base scenes, as well as on the outcomes, characterized by broad distribution and high difficulty assessment.|non-responsive|
|Is extension to three-dimensional space feasible?|We affirmed the feasibility of such an extension and proposed a concrete approach for its implementation.|non-responsive|
|Some errors in our manuscript|The recommended change has been implemented.|non-responsive|

### aj3h (initial score: 4, later score: 6)
|Questions|Response|Addressed or Not?|
|-|-|-|
|Lack manual validation in the benchmark|Additional experiments were performed and included in the revised manuscript.|Reviewer replied: issue addressed|
|Lack evaluation results on advanced mathematical reasoning agents|The CodePlot agent was selected to perform additional experiments, the results of which have been incorporated into the revised manuscript.|Reviewer replied: issue addressed|
|How to guarantee the reliability of the reasoning graphs underlying the synthetic tasks and the correctness in the generated questions?|We have provided a detailed explanation of the rules and procedures governing question generation, which has also been added to the revised article.|Reviewer replied: issue addressed|

### sFVL (initial score: 8)
|Questions|Response|Addressed or Not?|
|-|-|-|
|Lack a perturbed test set for evaluation|We have constructed a perturbed test set by applying random minor rotations to the images, and have incorporated the corresponding experimental results into the revised manuscript. We also state our commitment to extending this research to a broader range of real-world scenarios involving other forms of uncertainty in future work.|Reviewer replied: issue addressed|
|Recommend the inclusion of more multimodal large models|we have integrated Molmo into the evaluation and added the experimental findings to the revised version.|The reviewer responded and highlighted the significant value of our experimental results.|

---

> ### Author Response · Authors · 2025-11-30
> **Summary of Rebuttal during ICLR Incident (Part 2 of 2)**
>
> ### YWp2 (initial score: 4, later score: 6)
> |Questions|Response|Addressed or Not?|
> |-|-|-|
> |Raise concerns regarding the potential limitation of data distribution breadth due to the benchmark's heavy reliance on synthetic data, as well as the uncertainty in how well it reflects the difficulty level of real-world scenarios|We provided a detailed comparative analysis between GeoBench and real-world datasets in terms of distribution breadth and task difficulty, demonstrating from multiple perspectives that our dataset ensures sufficient diversity and complexity.|Reviewer replied: issue addressed|
> |Lack a well-established human baseline|Additional experiments were performed and included in the revised manuscript.|Reviewer replied: issue addressed|
> |Lack confidence testing|We conducted replicate experiments and computed confidence intervals, the results of which have been incorporated into the revised manuscript.|Reviewer replied: issue addressed|
> |Some inaccuracies in the textual descriptions|We have sincerely accepted the suggestions and made corresponding revisions in the paper.|Reviewer replied: issue addressed|
> |Question about the distinction and relationship between GeoBench and GeoBench-Solving|We have provided comprehensive explanations.|Reviewer replied: issue addressed
> |Suggestion to merge Tables 4 and 5 due to suspected high overlap|we explained that consolidating them was not adopted as it would result in overly dense data presentation and compromise visual clarity.|Reviewer replied: issue addressed|
> |Lack a citation for TrustGeoGen|We have made corrections.|Reviewer replied: issue addressed|
> |Lack specified experimental parameters|We have added parameter configuration.|Reviewer replied: issue addressed|
> |Ask about the design principles for distractors|We have elaborated on in the revised version.|Reviewer replied: issue addressed|
> |Is textual descriptions alone might lead to better model performance than images?|We have conducted corresponding ablation experiments. The results were added to the revised manuscript.|Reviewer replied: issue addressed|
> |Is the significant performance gap between N.P. and S.P. tasks is counterintuitive?|We have provided comprehensive explanations.|Reviewer replied: issue addressed|
> |Question about the methodology for generating feature vectors in Section 4.3|We have provided comprehensive explanations.|Reviewer replied: issue addressed|
> |Question about the criteria and implications of the two out-of-distribution benchmarks|We have provided comprehensive explanations.|Reviewer replied: issue addressed|
> |Request for a comparative example of Chain-of-Thought (CoT) enabled versus disabled settings|We have provided concrete case comparisons along with corresponding analysis.|Reviewer replied: issue addressed
>
> ***We have updated the revised manuscript based on the reviewers' comments.***
>
> ## Score Changes
>
> Our paper initially received scores of 4, 4, 6, and 8. Based on the reviewers' responses, the scores had likely been updated to 6, 6, 6, and 8 prior to the cut-off. Three reviewers (aj3h, sFVL, YWp2) acknowledged that their concerns were fully addressed. Among them, two (aj3h, YWp2) increased their scores from 4 to 6, while one (sFVL) maintained a positive score of 8. The remaining reviewer (FDHN), who initially gave a positive score of 6, has not yet responded. Nevertheless, we have provided a point-by-point response to the comments and believe this adequately addresses the reviewer's concerns.
>
> ## Final Claim
>
> The occurrence of the incident was highly unforeseen. We fully recognize the challenges faced by the ICLR organizing committee and the additional workload imposed on the newly assigned Area Chairs. We affirm that we have strictly adhered to all double-blind review regulations. It is our hope that the summary we have provided may help alleviate the your workload. We respectfully request that you consider accepting our work, and we remain committed to promoting fairness and progress within the academic community.

---

### Meta-Review · Area_Chair_DLyy · 2026-01-08

**Summary:**

The paper introduces GeoBench, a hierarchical benchmark designed to evaluate geometric reasoning for multimodal large language models, and presents extensive experimental results and analysis. Reviewers had concerns about how to verify the quality and legitimacy of the generated tasks, which the authors properly addressed by providing additional verification results. Overall, reviewers appreciate the effort behind this work.

**Reviewer Concerns:**

- How the automatically generated tasks are verified and controlled in terms of quality, as mentioned by many reviewers. The authors properly addressed this during the rebuttal.
- Significance tests, confidence intervals, or bootstrap estimates are necessary. The authors address this by providing confidence interval results.
- (Not addressed) How the automatically generated problems are distributed in terms of logical and reasoning complexity.

**Reviewer Scores:**

I believe Reviewer FDHN and Reviewer sFVL will maintain the score, Reviewer aj3h and Reviewer YWp2 will increase the score to 6.

---

### Decision · Program_Chairs · 2026-01-26

Accept (Poster)